# Electrically tunable collective motion of dissipative solitons in chiral nematic films

Yuan Shen [1] & Ingo Dierking [1✉]

From the motion of fish and birds, to migrating herds of ungulates, collective motion has attracted people for centuries. Active soft matter exhibits a plethora of emergent dynamic behaviors that mimic those of biological systems. Here we introduce an active system composed of dynamic dissipative solitons, i.e. directrons, which mimics the collective motion of living systems. Although the directrons are inanimate, artificial particle-like solitonic field configurations, they locally align their motions like their biological counterparts. Driven by external electric fields, hundreds of directrons are generated in a chiral nematic film. They start with random motions but self-organize into flocks and synchronize their motions. The directron flocks exhibit rich dynamic behaviors and induce population density fluctuations far larger than those in thermal equilibrium systems. They exhibit "turbulent" swimming patterns manifested by transient vortices and jets. They even distinguish topological defects, heading towards defects of positive topological strength and avoiding negative ones.

[1] Department of Physics and Astronomy, School of Natural Sciences, University of Manchester, Oxford Road, Manchester M13 9PL, UK.
✉email: ingo.dierking@manchester.ac.uk

Collective motion of animals is one of the most fascinating phenomena in our daily life. In spite of discrepancies in the size scales and the cognitive abilities of constituent individual components, systems such as schools of fish, flocks of birds, insect swarms, vertebrate herds and even human crowds produce analogous motion patterns with extended spatiotemporal coherence, indicating underlying universal principles[1]. Understanding these principles is not only interesting to physicists and biologists, but also vital to solving problems such as spreading of diseases, risk prevention at mass events, traffic jams, ecological environment problems, etc. However, it is difficult to study collective behavior by directly performing quantitative measurements in conventional macroscopic systems such as mammal herds, where tracking individual motions of a large population over long periods of time is extremely challenging. As an alternative, great achievement has been made through numerical modeling[2–4]. Moreover, different kinds of micro-scale experimental systems have also been developed to be utilized as active models for studying collective behavior[5–7].

Active soft matter, in which constituent building elements convert ambient free energy into mechanical work, is becoming increasingly recognized as a promising system for studying various out-of-equilibrium phenomena[8]. It shows rich emergent dynamic behaviors that are inaccessible to systems at thermal equilibrium. Recent works show that active systems, such as bacteria systems[9] and active colloidal systems[10], can serve as promising alternatives to conventional macroscopic systems to investigate the general behavior of collective motion. However, such systems are generally difficult to prepare and control. The bacteria may be harmful to the human body, and the synthesis of active colloidal particles is usually complex. Very recently, Sohn et al. reported schools of skyrmions, particle-like two-dimensional (2D) topological solitons[11]. However, only limited types of emergent collective behavior are realized because the individual skyrmions immediately synchronize their motions within seconds at a scale of the whole sample.

Solitons are self-sustained localized packets of waves that propagate in nonlinear media without changing shape, such as nerve pulses in living beings. They were first observed as water waves in a shallow canal by John Scott Russell in 1834[12], but their significance was not widely appreciated until 1965, when the word "soliton" was coined by Zabusky and Kruskal[13]. Nowadays, solitons have been investigated in many areas of physics, such as nonlinear photonics[14], Bose-Einstein condensates[15], superconductors[16], and magnetic materials[17], just to name a few. However, creating multidimensional solitons is still a great challenge in many physical systems due to their instability[18]. Liquid crystals (LCs) have been an ideal testbed for studying solitons for decades[19]. Different kinds of solitons have been produced in LCs[19], both immobile[20–24] and mobile ones[11,25–28]. Recently, electrically driven 3D dissipative solitons coined as "director bullets" or "directrons" have received increasing attention[29–35]. These directrons were firstly reported by Brand et al. in 1997, which were called "butterflies" by the authors, but did not receive great attention at that time[36]. The directrons represent nonsingular director perturbations that propagate through a uniform nematic bulk without losing their identities. Within the directrons, the director field oscillates with the frequency of the applied electric field due to the flexoelectric effect, breaking the mirror symmetry of the directrons and driving them to move.

Here we show an emergent collective motion of such directrons. Individual directrons consume electric energy and convert it into mechanical motions. Depending on the applied voltage, they exhibit different dynamic behaviors. At low voltages, the directrons self-organize into chains, loops, flocks, and swarms that move coherently like, for example, schools of fish (Fig. 1a, b). At high voltages, the density of the directrons increases dramatically. Tens of thousands of directrons start from random orientations and motions but then synchronize their motions through collisions and short-range interactions with each other and develop ferromagnetic-like order within tens of seconds, as described by the well-known "Vicsek model"[3]. Further increasing the voltage leads to a dynamic transition of the collective motion from a wavy motion regime to linear motion and eventually ends into chaotic, incoherent motion. The directrons also exhibit an emergent "turbulent" swimming pattern, which is manifested by recurring transient vortices and jets. Furthermore, we show that the collective motion of the directrons can be controlled by topological defects, where directrons swirl around defects of positive topological strength and avoid the ones of negative strength, in accordance with the behavior of bacteria reported before by Lavrentovich and co-workers[37]. Our findings show that active soft matter formed by directrons exhibits rich emergent collective dynamic behaviors that mimic those of living systems and provide a tunable model for studying collective motion.

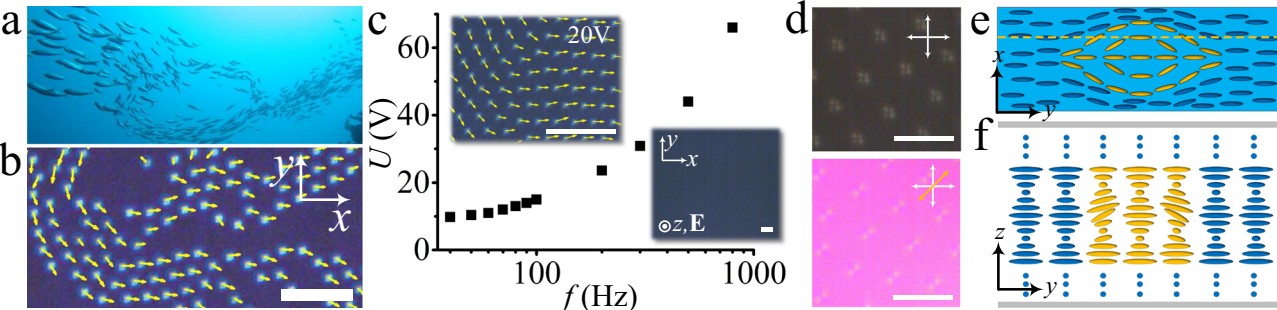

**Fig. 1 Collective motion of directrons. a** Photograph of schools of fish. Figure from Joanna Penn, Flickr: https://www.flickr.com/photos/38314728@N08/3997721496/in/dateposted/ [reproduced by permission]. **b** Polarizing micrograph of collectively moving directrons with their velocities indicated as yellow arrows. $U = 16$ V, $f = 100$ Hz. Scale bar 50 μm. The polarizers are parallel to the $x$- and $y$-axes, respectively. **c** Frequency dependence of the threshold for generation of directrons. The insets show the micrographs below (0 V) and above the thresholds ($U = 20$ V, $f = 100$ Hz) of the generation of the directrons. The yellow arrows represent the velocities of the directrons. Scale bars 50 μm. **d** The polarizing micrographs of directrons at $f = 100$ Hz, $U = 20$ V. Scale bars 20 μm. The white crossed arrows represent the polarizers, and the yellow arrow represent the optical axis of the $\lambda$-plate. **e** The schematic director structure of a directron in the middle layer of the chiral nematic sample. **f** The schematic director structure of a directron in the $yz$ plane of the cross section along the dashed yellow line in **e**. The director field within the directron is represented as yellow ellipses and the homogeneous director field outside the directron is represented as blue ellipses. The top and bottom sections of the sample in **f** are homogeneously aligned helical structures and are represented by blue dots.

## Results

**Formation of directrons**. Collective motion, like it is exhibited by schools of fish (Fig. 1a), is accompanied with inhomogeneities and the formation of dynamic groups whose volume and shape change as the groups turn and arc but remain cohesive. Similar behavior is found in our rather unusual active soft matter system formed by hundreds and thousands of dynamic particle-like solitonic director field configurations within a cholesteric LC film (Fig. 1b), which exhibits complex, coordinated, spatiotemporal dynamical patterns. The experimental setup is similar to that in LC displays, where a thin film of cholesteric LC is sandwiched between two pieces of glass substrates which are coated with indium tin oxide (ITO) layers as transparent electrodes and rubbed polyimide layers as the planar alignment layers, i.e. the director near the glass substrates aligns along the rubbing direction (along the $x$-axis, Fig. 1b). Along the normal to the glass substrates, the director twists continuously along a helical axis at a constant rate. The cell gap $d \sim 9.5\,\mu m$. An alternating-current (AC) electric field, **E**, is applied perpendicular to the LC film (along the $z$-axis). The directrons emerge when the applied voltage exceeds a frequency-dependent threshold (Fig. 1c). The formation mechanism of the directrons is not completely understood yet and requires further experimental and theoretical investigations. Recent publications by Pikin suggested that the generation of the directrons may be due to the interaction of injected electron clouds with nematic molecules[38,39]. However, the charge injection usually happens under a DC electric field or AC electric field with low frequencies and is suppressed as soon as the oscillation frequency of the applied field exceeds a few cycles[40]. Although, the directrons here exist at relatively high frequencies (up to 800 Hz, Fig. 1c), one cannot exclude the possibility of charge injection since it may still occur at high voltages. To totally exclude the possibility of charge injection, further experiments are required, for instance, one can make a cell with "blocking" electrodes[41]. On the other hand, according to previous investigations[29–31,34,35], the formation of the directrons is closely dependent on the dielectric and conductivity anisotropies of the LC materials. the directrons only occur in the limited range of moderate conductivity. If the conductivity is higher than the range, only global electro-convective patterns are observed[34]. The conductivity anisotropy of the LC in the present study is $\Delta\sigma = 1.3 \times 10^{-8}\,\Omega^{-1}m^{-1}$ (Methods), which is relatively small compared to the ones used in studies of electro-convections ($\sim 10^{-7}\,\Omega^{-1}m^{-1}$)[42,43]. The significance of this small conductivity anisotropy can be understood by considering the coupling between the electric field and the space charges. According to the Carr–Helfrich electro-convection mechanism[44,45], the positive conductivity anisotropy and the bend fluctuation in a nematic induce ion segregation and form space charges which are high and uniformly distributed in space. These space charges produce transverse Coulomb forces which offset the normal elastic and dielectric torques and cause instability, usually in the form of space-filling periodic stripes. However, due to the relatively low conductivity, there is not sufficient charge accumulation to produce a strong enough dielectric torque to induce a global electro-convection pattern. Instead, the director field around the space charges is locally deformed, inducing a flexoelectric polarization. As a result, the director within the local deformations oscillates with the frequency of the applied electric field[29–31], leading to the formation of the directrons.

**Dynamics of directrons**. Unlike the directrons in achiral nematics reported previously, which move either parallel or perpendicular to the alignment direction[29,31], the directrons here travel in random directions in the $xy$ plane independent of the alignment direction (Fig. 2a). Such a behavior may be attributed to the rotational symmetry of the small pitch helical structure of the LC system, which suppresses the influence of the rubbing alignment. Moreover, the director deformation of the directrons reaches maximum in the middle layer of the LC media, and gradually diminishes as one moves toward the top and bottom cell substrates[29], which further reduces the influence of the rubbing alignment. This can be demonstrated by the isotropic Brownian diffusion of a colloidal micro-particle in our system (Supplementary Fig. 1), which always exhibits an anisotropic diffusion behavior in achiral nematics[46]. The structure of the directrons is deduced from the polarizing micrograph, which shows a quadrupolar symmetry (Fig. 1d). Within the directron, the director deviates from the uniform state due to the transverse Coulomb forces provided by space charges as well as the flexoelectric polarization (Fig. 1e, f). Unlike the topological solitons, such as skyrmions[23], in which the director field cannot be continuously transformed into a uniform state, the directrons are topologically trivial. The director also oscillates, both in the $xy$-plane and out of the plane (Fig. 1e, f), with the frequency of the applied electric field due to the flexoelectric effect[29–31], which leads to the periodic deformation of the directron structure (Supplementary Fig. 2) and propels the directrons to move. The moving direction of the directrons is determined by the breaking of the quadrupolar symmetric structure (Fig. 2a insets), i.e. the directrons moving along the $x$-axis lack the left-right symmetry with respect to the $yz$ plane; vice versa, the directrons moving along the $y$-axis lack the fore-aft symmetry about the $xz$ plane[29–32]. One may suspect that there are some kind of fluid flows that carry the directron's motion. However, since the dielectric anisotropy of the LC material is negative ($\Delta\varepsilon = -4.6$), there is no backflow generated by the rotation of the director field. No electro-convection pattern is observed, which can exclude the electro-convective flows. The isotropic flows generated by the injection of ions usually occur at very low frequencies[40]. The only possibility left is the flow generated by ion motion. However, this flow also usually happens at relatively low frequencies, i.e., the conductive regime, which is limited from above by the critical frequency, $f_c$[47]. According to our calculation, $f_c \sim 128\,Hz$ in our system (*Methods*). However, the directrons can move effectively at relatively high frequencies (up to 800 Hz or even higher). Furthermore, particle tracking does not show obvious existence of flows (*Methods*). So, as far as we can conclude, it appears that there is no such kind of flows which can carry the directron's motion. The directrons can also switch their moving directions during motion due to the change of the asymmetry of their structures induced by, for instance, director fluctuations or collisions with other directrons and inhomogeneities, such as dust particles. It is found that the directrons cannot pass through each other like waves, as reported for those in achiral nematics[29–31]. Instead, the directrons attract each other at relatively long ranges but repel at short distances (Fig. 2e). Since the directrons are nonsingular director deformations, such an interaction is more likely to stem from the elastic distortion of the director field. Figure 2b shows that two directrons move in opposite directions collide and repel each other into different directions like true particles, but then attract each other and form a linear chain that moves coherently along a spontaneously chosen direction (Supplementary Movie 1). If more directrons get close to each other, they can not only form linear chains but also form closed loops (where the head and the tail of the linear chains are connected) which rotate continuously in the $xy$ plane about their rotation axes along the $z$-axis (Fig. 2c, d and Supplementary Fig. 3 and Supplementary Movie 2). To probe how the dynamics of directrons changes with increasing number density, we measure the displacement of directron flocks of different sizes (flocks

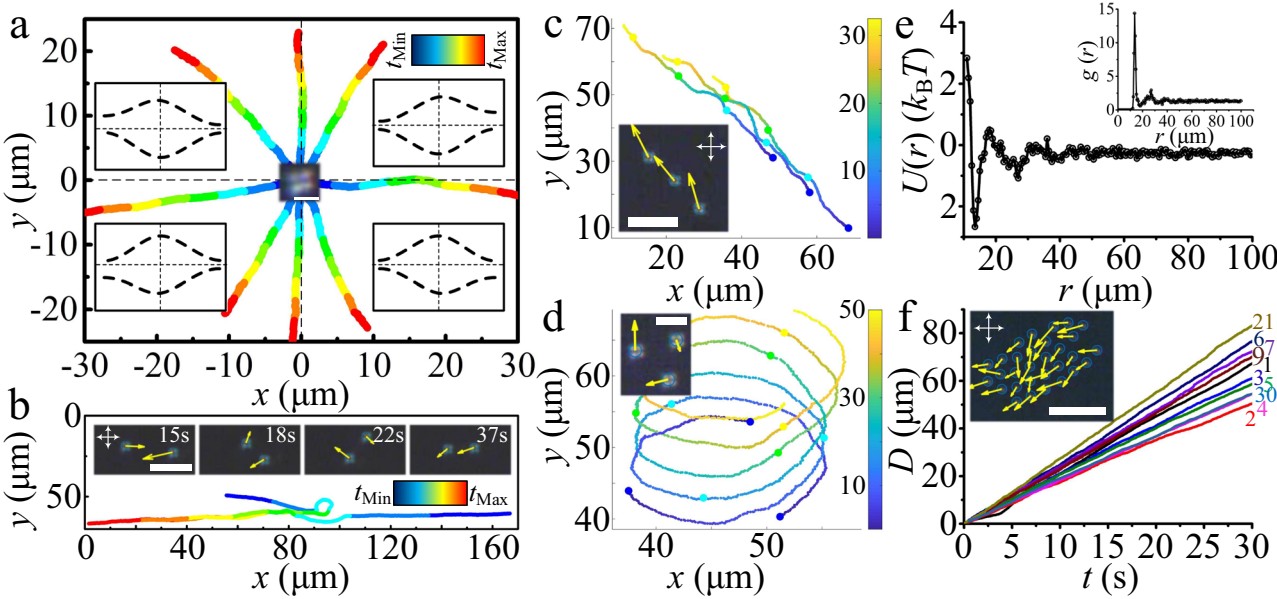

**Fig. 2 Dynamics of directrons. a** Trajectories of eight individual directrons at $U = 15.4$ V, $f = 100$ Hz, colored with time corresponding to the color bar ($t_{Min} = 0$ s, $t_{Max} = 10$ s). The insets in four quadrants show the schematic symmetry-breaking middle-layer director structures of the directrons. The inset in the middle shows the polarizing micrograph of a directron. Scale bar 5 μm. **b** Trajectories of two directrons at $U = 15.4$ V, $f = 100$ Hz colliding with each other, colored with time corresponding to the color bar ($t_{Min} = 0$ s, $t_{Max} = 67$ s). The insets show the time sequences of micrographs of the two directrons with yellow arrows representing the velocities, scale bar 20 μm. **c, d** Trajectories of three directrons at $U = 15.4$ V, $f = 100$ Hz, colored with time corresponding to the color bars (unit (s)). The directrons demonstrate linear motion (**c**) and circular motion (**d**). The insets show the micrographs of the three directrons with their velocities represented as yellow arrows. Scale bars 20 μm in **c** and 10 μm in **d**. **e** Pair interaction potential function (extracted from the radial distribution function $g(r)$ shown in the inset) of the directrons at $U = 15.2$ V, $f = 100$ Hz. **f** Dependence of displacements ($D$) of groups of different number of directrons on time ($t$). The inset shows the micrograph of the group composed of 30 directrons with their velocities representing as yellow arrows. Scale bar 50 μm. The white crossed arrows in the micrographs indicate the polarizers.

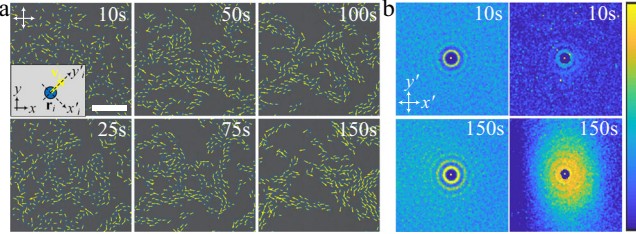

**Fig. 3 Formation of directron flocks. a** Polarizing micrographs of directrons ($U = 16.2$ V, $f = 100$ Hz.) with their velocities represented as yellow arrows at different times after applying the electric field. Scale bar 100 μm. The crossed polarizers are indicated as white arrows. The time after the application of the electric field is indicated. A laboratory coordinate frame ($x$ $y$) and a local coordinate frame ($x'$ $y'$) are defined in the inset. **b** 2D radial distribution functions ($g$) (first column) and 2D spatial velocity correlation functions ($g_v$) (second column) of the directrons colored according to the color bar at 10 s and 150 s after applying the electric field, respectively. The color bar changes linearly from 0 (dark blue) to 4 (light yellow) for $g$ and from 0 (dark blue) to 1 (light yellow) for $g_v$. Image sizes represent an area of 200 μm × 200 μm. The 2D spatial velocity correlation functions are averaged over 5 s (ten frames).

composed of different number of directrons, $n$) within a specific duration, but no obvious dependence is observed (Fig. 2f).

**Flocks of directrons.** Hundreds of directrons emerge after increasing the voltage over the critical value (Fig. 1c). At relatively low voltages, they form a gas-like phase in which they are distributed sparsely throughout the whole sample and move in random directions, occasionally experiencing collisions,

depending on the directron density. Due to the long-range attractive pairwise interaction, the directrons gradually travel towards each other and form dynamic flocks of different sizes which move in different directions (Supplementary Movie 3). Within each flock, the directrons move coherently on average in the same direction (Fig. 3a). To quantify the correlation between the directrons, we compute spatial correlation functions in a "local coordinate frame" with the $y$-axis ($y_i'$) along the velocity vector of the $i^{th}$ directron ($\mathbf{v}_i$) and the $x$-axis ($x_i'$) perpendicular to it (Fig. 3a inset). The radial distribution function, $g(r)$, quantifies the probability of finding another directron in a unit area at the point ($x,y$) away from the reference directron, where $r$ represents the distance between directrons. It shows an excluded volume with a radius of ~10 μm due to the short-range repulsive interaction and a peak at $r \sim 13$ μm (Fig. 3b), which is consistent with the pairwise interaction between directrons (Fig. 2e). Close neighbors show similar velocity to the reference directron as evidenced by the peak corresponding to the first nearest neighbor of the spatial velocity correlation function, $g_v(r)$. The peak corresponding to the second nearest neighbor of $g_v(r)$ is indiscernible at $t = 10$ s, indicating the short-range velocity correlation between directrons at an early stage. With motion progressing, both $g(r)$ and $g_v(r)$ become increasingly long-range with an increased number and value of peaks, indicating the formation of flocks (Fig. 3b). The spatial velocity correlation function at $t = 150$ s is anisotropic with respect to the direction of collective motion, which shows a longer correlation along the $y'$-axis (Fig. 3b). Such an anisotropy of the spatial correlation functions has been identified as one of the key properties of collective motion[48]. It has also been observed in other active systems, such as bacteria suspensions[9] and driven filament systems[49], which stems from the anisotropic shape of the individual components. However, the

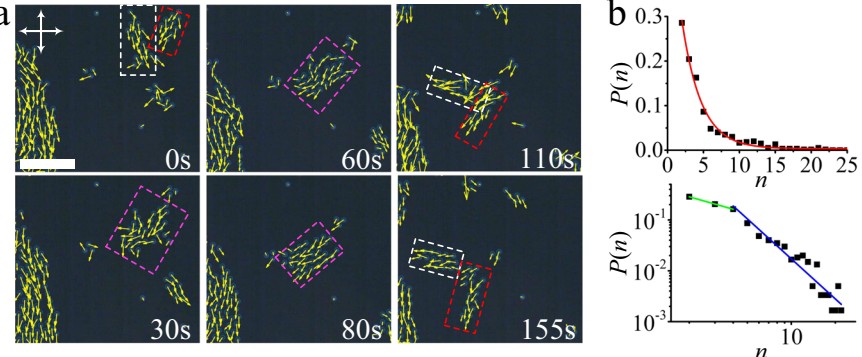

**Fig. 4 Directron flocks. a** Polarizing micrographs of fusion and fission of directron flocks at different times. Directron flocks are indicated by dashed squares with different colors. Scale bar 100 μm. The crossed polarizers are indicated as white arrows. $U = 15.4$ V, $f = 100$ Hz. **b** Frequency distribution of directron flock sizes (number of individual directrons, $n$, in each flock) at $U = 15.5$ V, $f = 100$ Hz. The top figure demonstrates the probability of finding a directron flock composed of $n$ directrons. The red line represents the exponential fit of the experimental data (black squares). The bottom figure shows the log-log plot of the distribution. The green and blue lines are linear fits of the experimental data with slopes $\beta_1 = -0.8$ and $\beta_2 = -2.6$, respectively.

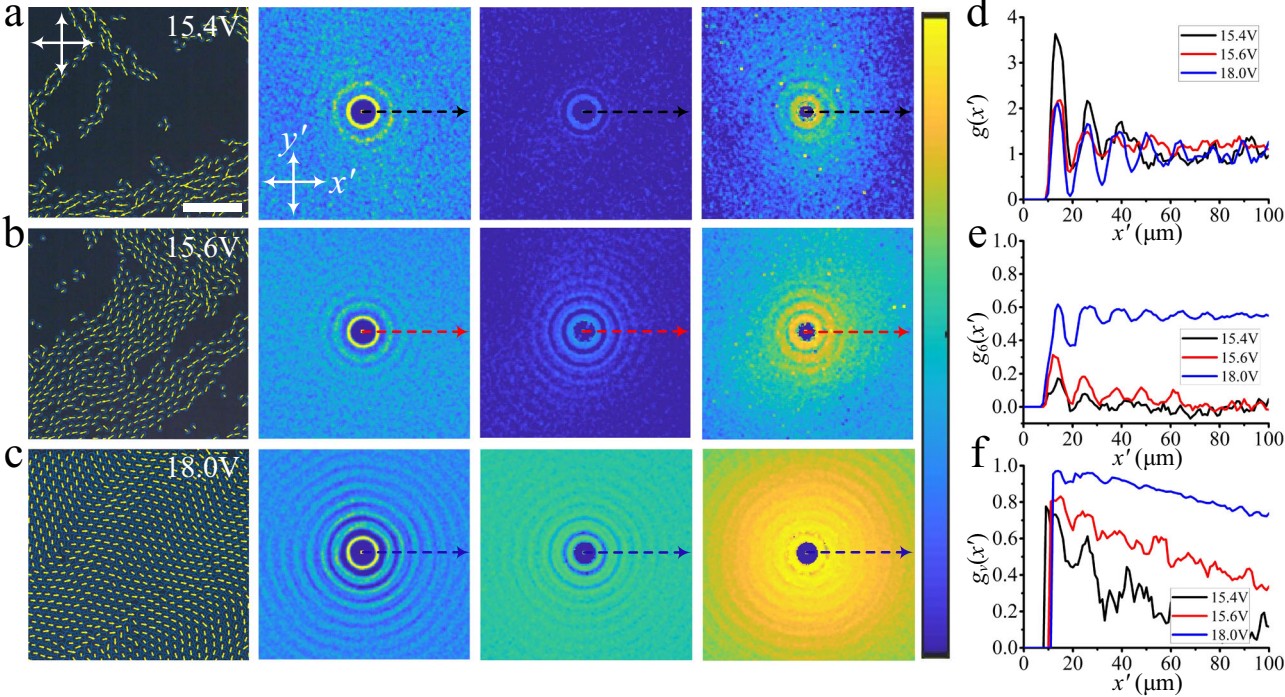

**Fig. 5 Collective motion of directrons at different voltages.** Micrographs (first column, scale bar 100 μm), 2D radial distribution functions ($g$, second column), 2D hexatic bond orientational correlation functions ($g_6$, third column) and 2D spatial velocity correlation functions ($g_v$, forth column) of the directrons at different voltages, (**a**) $U = 15.4$ V, (**b**) $U = 15.6$ V, and (**c**) $U = 18.0$ V. The frequency is fixed at $f = 100$ Hz. The image sizes of the 2D correlation functions represent an area of 200 μm × 200 μm. The color bar changes linearly from 0 (dark blue) to 3 (light yellow) for $g$, and from 0 (dark blue) to 1 (light yellow) for $g_6$ and $g_v$. The transverse profiles (as indicated by the dashed arrows) of the corresponding (**d**) radial distribution functions ($g(x')$), (**e**) bond orientational correlation functions ($g_6(x')$), and (**f**) spatial velocity correlation functions ($g_v(x')$). The spatial velocity correlation functions are averaged over 10 s (20 frames).

directrons here are basically circular in the $xy$ plane and the anisotropy of the velocity correlation function is due to the anisotropic shape of the directron flocks, which are relatively elongated along their moving directions (Fig. 3a). The radial distribution functions do not show obvious anisotropy due to the short-range pair correlation between the directrons. The directron flocks change their size through fusion and fission processes, as shown in Fig. 4a. Two directron flocks moving in different directions collide with each other and then combine into a larger flock. Right after the collision, the motions of the directrons within the flock become slightly disordered with a relatively large

velocity deviation, but they quickly synchronize their motions and move coherently in a specific direction. The large flock is not stable; after traveling a certain distance, it fragments into two smaller flocks which move away from each other in different directions (Supplementary Movie 4). Figure 4b shows the size distribution of directron flocks at a fixed electric field, which is exponentially distributed. Such an exponential distribution is consistent with the group size distribution of fish schools as reported by Flierl et al.[50], and predicted by Okubo[51]. However, Niwa argued that the group size distributions fit a truncated power law with a crossover to an exponential decay and is

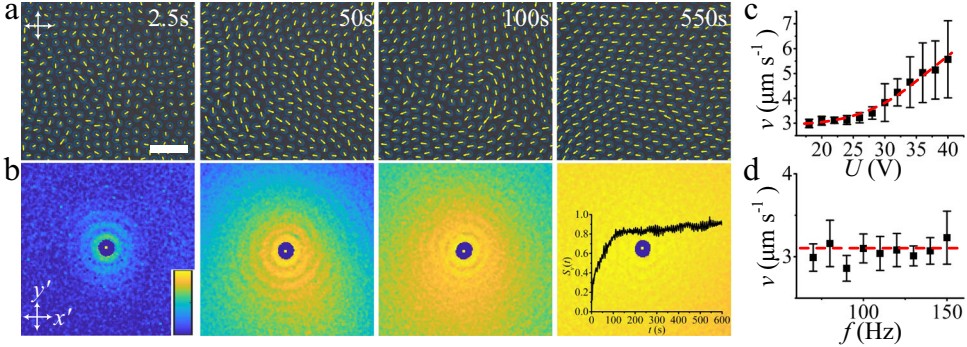

**Fig. 6 Temporal evolution of directron velocities and their dependence on electric fields. a** Polarizing micrographs of directrons at $U = 20$ V, $f = 100$ Hz, with their velocities represented as yellow arrows. Scale bar 50 μm. The crossed polarizers are indicated as white arrows. The time after the application of the electric field is indicated in each image. **b** 2D spatial velocity correlation functions corresponding to **a** colored according to the color bar which changes linearly from 0 (dark blue) to 1 (light yellow). Image sizes represent an area of 200 μm × 200 μm. The inset shows the temporal evolution of the velocity order parameter, $S_v(t)$. The 2D spatial velocity correlation functions are averaged over 5 s (10 frames). **c** The dependence of directron velocity on voltage at fixed frequency ($f = 100$ Hz). **d** The dependence of directron velocity on frequency at fixed voltage ($U = 20$ V). The error bars are calculated from the standard deviation of velocities of hundreds of different directrons.

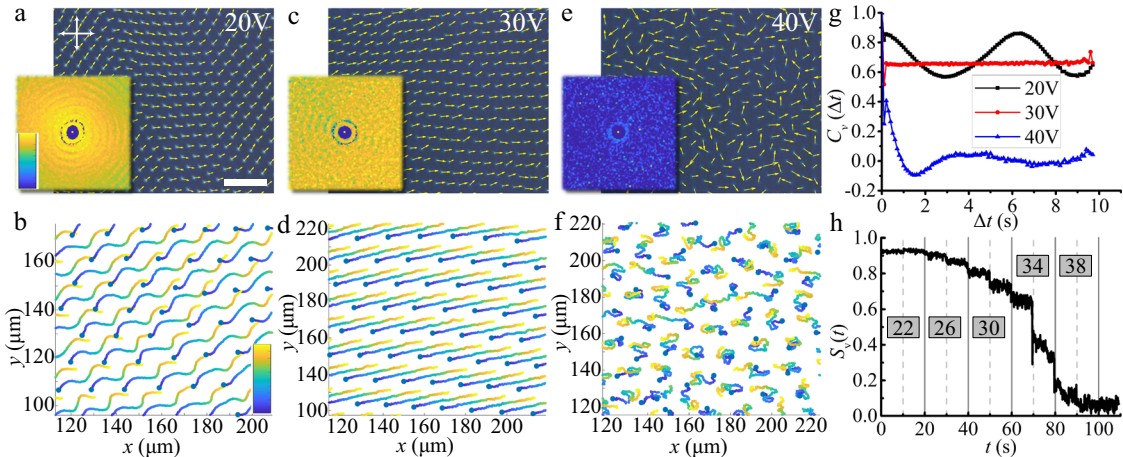

**Fig. 7 Voltage dependence of collective motion.** Polarizing micrographs of directrons at (**a**) $U = 20$ V, (**c**) $U = 30$ V, and (**e**) $U = 40$ V, respectively, with their velocities represented as yellow arrows. The frequency is fixed at $f = 100$ Hz. Scale bar 50 μm. The crossed polarizers are indicated as white arrows. The insets are the corresponding 2D spatial velocity correlation functions colored according to the color bar which changes linearly from 0 (dark blue) to 1 (light yellow). The image sizes represent an area of 200 μm × 200 μm. The 2D spatial velocity correlation functions are averaged over 10 s (20 frames). Trajectories of directrons at (**b**) $U = 20$ V, (**d**) $U = 30$ V, and (**f**) $U = 40$ V, respectively, colored with time corresponding to the color bar, which changes linearly from $t = 0$ s (dark blue) to $t = 10$ s (light yellow). **g** The temporal velocity correlation functions ($C_v(\Delta t)$) of directrons at different voltages corresponding to **a**, **c**, and **e**. **h** Time dependence of the velocity order parameter, $S_v$, with the applied voltage (indicated as numbers in gray squares) gradually increasing from 20 V to 40 V in a step of 2 V every 10 s.

dependent on the cutoff size[52,53]. If the cutoff size is small, the exponential decay will be the only part of the functionality, but if the cutoff size is large, one may predict that the group size distribution would better fit a power law[52,53]. We plot the size distribution of directron flocks on a log-log scale and find a cutoff size of $n = 4$ (Fig. 4b bottom).

**Electrically controllable collective dynamic behavior.** The density of the directrons depends on the applied voltage. By increasing the voltage, more and more directrons emerge. Figure 5 shows the dynamic steady states of the collectively moving directrons at varied voltages. It is found that by increasing the directron population density, the radial distribution function ($g$), hexatic bond orientational correlation function ($g_6$), and spatial velocity correlation function ($g_v$) become more and more long-range. This is because at low density, the directrons form small flocks which move incoherently in random directions. By

increasing the directron density, small flocks collide with each other and form larger flocks within which the directrons pack into a hexagonal structure and move coherently. At $U = 18$ V, the whole sample plane is filled with directrons which are orderly arranged due to the short-range repulsive interaction and on average move coherently in the same direction. The correlation lengths of the spatial correlation functions are of the order of tens or even hundreds of micrometers and are much larger than those of bacteria suspensions which are usually only several micrometers[9]. Such a difference may be due to the short-range repulsive and long-range attractive interaction between the directrons which is absent among bacteria. The spatial velocity correlation functions in Fig. 5b, c do not show obvious anisotropic profiles which is because the directrons do not form anisotropic flocks; instead they form large-scale or even global directron flows which extend hundreds and thousands of micrometers due to the high directron density. At high packing fractions, the applied **E** initially induces random motion of

directrons. Unlike the behavior at relatively low population density where dynamic flocks of directrons form with time (Fig. 3), the directrons here collide and interact with each other, leading to a large-scale emergent coherent directional motion (Fig. 6a and Supplementary Movie 5). This behavior arises from many-body interactions between directrons as predicted by the "Vicsek model" where active, point-like particles interact so that they tend to align their velocities with those of their neighbors. The velocity order parameter, $S_v$, which characterizes the degree of ordering of directron velocities, gradually increases from ~0.1 to ~0.8 within tens of seconds and saturates at $S_v \sim 0.9$ (Fig. 6b inset), indicating the emergence of coherent unidirectional motion of directrons. As a result, the spatial velocity correlation function becomes more and more long-range (Fig. 6b). Such a phenomenon is usually seen in everyday life as one walks towards a group of pigeons. The pigeons are startled and take off en masse. Initially, they fly in random directions, but soon the flock orders and moves away coherently.

The velocity of the directrons is also dependent on the electric field. The amplitude of the directron velocity, $v$, increases with increasing applied voltage but shows no obvious dependence on the frequency of the electric field (Fig. 6c, d). At the same time, by tuning the applied voltage, the directrons show different collective dynamic behaviors (Fig. 7 and Supplementary Movie 6). It is observed that the collective motion of directrons gradually transforms from a wavy motion (Fig. 7a, b) to a linear motion (Fig. 7c, d) with increasing voltage. Although, the spatial velocity correlation functions at $U = 20\,V$ and $U = 30\,V$ are similar to each other, which show strong long-range correlations (insets in Fig. 7a, c), the temporal velocity correlation functions ($C_v(\Delta t)$) are very different from each other. The $C_v(\Delta t)$ at $U = 20\,V$ changes periodically and forms a sinusoidal wave, which is due to the wavy motion of the directrons (Fig. 7g). At $U = 30\,V$, the $C_v(\Delta t)$ keeps constant which is in accordance with the linear motion of the directrons (Fig. 7g). At $U = 40\,V$ (Fig. 7e, f), a chaotic incoherent motion is observed where the directrons move randomly with only weak velocity correlation at very short ranges (Fig. 7e inset). The $C_v(\Delta t)$ firstly decays to zero within 1 s and then develops a negative correlation with a minimum at $\Delta t \sim 1.5\,s$, which then gradually increases to 0 (Fig. 7g). The temporal velocity correlation functions at varied voltages all show a two-step relaxation. The first relaxation always occurs immediately within 0.1 s, which is barely resolved. Such a two-step relaxation has also been observed in bacteria suspensions[54]. The velocity order parameter, $S_v$, gradually decreases with increasing voltage at first, but then suddenly decreases from ~0.7 to ~0.4 at $U = 34\,V$ and eventually falls to 0 at $U = 38\,V$ (Fig. 7h). Such an order-disorder transition is due to the increase of the background noise of the system[3], which can be caused by different reasons, such as the increase of the velocity of the directrons, the distortion of the directron structures, the ion injection, the hydrodynamic flow induced by the motion of directrons, etc. Such a phenomenon is also observed in other active systems, such as granular systems, in which the systems show an order-disorder transition by increasing the amplitude of the vibration of the systems[55].

Directrons in flocks are packed closely, which causes high local density. At the same time, these flocks are mobile and they usually leave empty space in regions they just travel through, leading to low density in those regions. As a result, dynamic flocks produce large density fluctuations, as shown by the time evolution of the total number of directrons, $N$, in the whole viewing area in Fig. 8b. This measurement shows a maximum about 1.4 times as large as the minimum and a standard deviation of $\Delta N \sim 31$, which is about 7.6% of the mean $\langle N \rangle = 406$. Apart from the large magnitude, the standard deviations scale with the

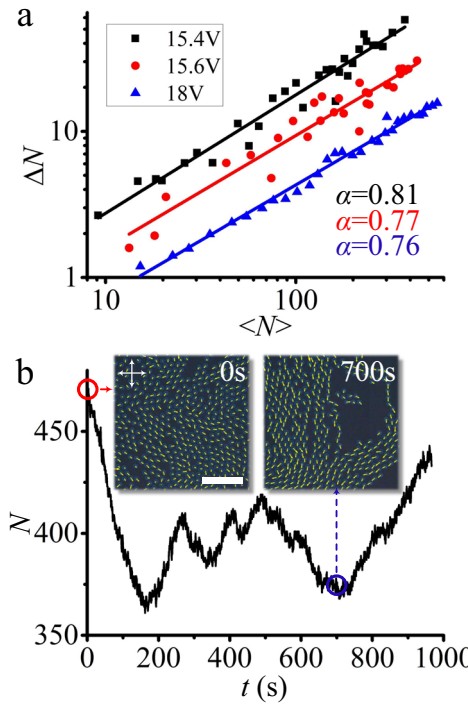

**Fig. 8 Anomalous density fluctuations in collectively moving directrons.**
**a** The log-log plot of the standard deviation $\Delta N$ versus the mean directron number $\langle N \rangle$ for three different voltages: $U = 15.4\,V$ (black squares), $U = 15.6\,V$ (red circles) and $U = 18.0\,V$ (blue triangles). The frequency is fixed at $f = 100\,Hz$. Solid lines are the linear fits of the experimental data. **b** Total number of directrons, $N$, in the field of view (306 μm * 306 μm) as a function of time. The insets show the polarizing micrographs of the directrons at $t = 0\,s$ (red circle) and $t = 700\,s$ (blue circle), respectively, with their velocities represented as yellow arrows. $U = 15.6\,V$, $f = 100\,Hz$. Scale bar 100 μm. The crossed polarizers are indicated as white arrows.

means differently from those in thermodynamic equilibrium systems, where fluctuations obey the central limit theorem, $\Delta N \propto \langle N \rangle^{1/2}$[56–58]. In Fig. 8a, it is found that directrons in flocks exhibit anomalous density fluctuations, also called giant number fluctuations[56]. For three different directron densities (voltages), the standard deviation $\Delta N$ grows more rapidly than $\langle N \rangle^{1/2}$ and scales as $\Delta N \propto \langle N \rangle^{\alpha}$, where $\alpha \sim 0.785 \pm 0.025$. Such giant number fluctuations have been reported in numerical simulations of self-propelled polar particles where $\alpha = 0.8$ is found[57] as well as in collective motion of bacteria where $\alpha = 0.75$ is reported[9], which are close to the values measured for directrons here.

**Recurring transient vortices and jets**. The emergence of transient vortices and jets in ensembles of self-propelling agents has been observed in various active systems, such as swarming bacteria[54], vibrating granular particles[59], and active colloidal systems[60]. We show that similar patterns can also be generated in our system. Here, the same LC material is filled into a homogeneous cell with a larger cell gap ($d \sim 19.7$ μm). By increasing the applied voltage to some specific values, hundreds and thousands of directrons emerge and exhibit a "turbulent" swimming pattern manifested by recurring transient vortices and jets (Fig. 9). The size of the directrons is slightly larger than the ones mentioned above (Supplementary Fig. 4). The instantaneous velocity field of the directrons in Fig. 9 reveal intense vortices with the length scales over hundreds of micrometers which are significantly exceeding the size of individual directions. The vortices spontaneously form from time to time throughout the sample. However,

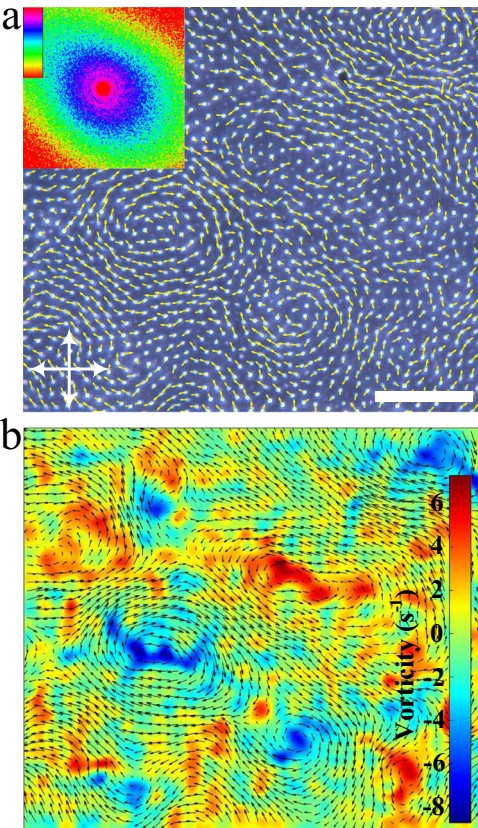

**Fig. 9 Instantaneous velocity field of directrons. a** Polarizing microscopy of directrons with their velocities represented as small yellow arrows. $U = 100$ V, $f = 500$ Hz. The crossed white arrows indicate the polarizers. Scale bar 200 μm. The inset represents the corresponding 2D spatial velocity correlation function colored according to the color bar which changes linearly from −0.2 to 1. The image size of the 2D velocity spatial correlation function represent an area of 400 μm × 400 μm. The 2D spatial velocity correlation functions are averaged over ~1.4 s (20 frames). **b** Color map of vortices. The bar shows a linear scale of vorticity. The black arrows show the PIV flow field. .

they are not stable as they gradually move through the nematic bulk and disappear within seconds (Supplementary Movie 7). The directrons show strong short-range but weak long-range velocity correlation during motion (Fig. 9a inset). The spatial velocity correlation function ($g_v(r)$) also develops an anisotropic profile and shows strong negative correlations at distances $r \sim 200$ μm, which corresponds to the characteristic length scale of the vortices. Such negative velocity correlations are due to the formation of the vortices and have also been reported in turbulent bacteria suspensions[54] and active colloidal systems[61]. The temporal velocity correlation function ($C_v(\Delta t)$) also shows a two-step relaxation (Supplementary Fig. 5) like the ones in Fig. 7g, which firstly decreases to ~0.75 within 0.1 s, and then gradually decays to 0. The second relaxation demonstrates the process of the disintegration of coherent structures, which decays slowly within seconds (Supplementary Fig. 5). Figure 9b shows the instantaneous distribution and amplitude of the vortices obtained through particle image velocimetry (PIV). The formation of such a "turbulent" swimming pattern in cells with larger cell gap can possibly be attributed to the larger space between the directrons and the larger velocity of the directrons, which induce complicated hydrodynamic interactions between directrons. More details about such a turbulent collective behavior will be reported elsewhere.

**Circular collective motion of directrons commanded by topological defects**. It was reported that bacteria can sense the topological strength of defects and move around defects of a positive topological strength ($s > 0$) and avoid negative strength defects ($s < 0$)[37]. A similar phenomenon is also observed in this soliton system. Here, the LC sample is confined between two substrates with homeotropic anchoring conditions (cell gap $d \sim 9.4$ μm). It should be noted that the LC material used here has a pitch ($p = 10$ μm) larger than the ones used in previous experiments ($p = 2$ μm). The reason that we choose a larger pitch here is because there will be a lot of complicated topological defects and disclinations if the pitch is small, which will trap and hinder the motion of directrons. By applying an electric field normal to the cell substrates, due to the negative dielectric anisotropy ($\Delta \varepsilon < 0$) of the LC media, the director field is reoriented into the $xy$ plane parallel to the cell substrates, leading to the formation of umbilic defects with topological strength, $s = \pm 1$[62,63]. One may suggest that these defects are the dowser textures[64] since they look similar to the umbilic defects in polarizing microscopy. However, the core of the dowser texture is a hedgehog point defect whose size cannot be tuned by the electric field (although it can be transformed into a looped line defect)[65]. The size of the defect core observed here can be continuously changed by tuning the applied electric field (Supplementary Figs. 6 and 7) which is consistent with the property of umbilic defects whose structure is nonsingular[62]. At the same time, directrons are also generated throughout the sample. The director structure of the directrons observed here maybe slightly different from the ones described above due to the larger pitch. However, they are of the same type because they show similar formation, stability and dynamics. It is found that the directrons travel towards the defects with $s = +1$ and gather into circulating flocks which move coherently around the $s = +1$ cores. In contrast, the directrons deplete from the defects with $s = -1$, moving away from them (Fig. 10a, b and Supplementary Movie 8). Moreover, the pair of $s = \pm 1$ defects can also be generated by introducing micro-particles into the system, where the micro-particles act as the cores of the $s = +1$ defects and simultaneously induce an accompanied $s = -1$ defect nearby[63]. Similarly, the directrons swarm around the micro-particles and avoid the $s = -1$ defects (Fig. 10d, e and Supplementary Movie 9). By turning off the electric field, the directrons disappear immediately and the sample turns into a fingerprint texture, from which the topological strength of the defects can be deduced (Fig. 10c, f). Guiding the motion of solitons with umbilic defects was also recently reported by Sohn et al.[66]. However, unlike the directrons here which are trapped and persistently swirling around the $s = +1$ defect but repelled by the $s = -1$ defect, the solitons, (skyrmions, in their case), are deflected and sidetracked by the umbilic defects.

## Discussion

Collective behavior has been broadly studied in different kinds of systems, including bacteria suspensions[9], driven filaments[7], granular systems[59], and various active colloidal systems[10,60]. Recently, such a behavior is also realized with topological solitons in cholesteric LC systems by Sohn et al.[11,66]. In their studies, hundreds and thousands of topological solitons, skyrmions, are generated and driven into random motions by electric fields which immediately synchronize their motions within seconds and exhibit collective motions along spontaneously chosen directions. However, there are only limited types of collective behavior of this electrically driven active motion, and all skyrmions, even at very low packing fractions, tend to synchronize to move together at a constant velocity and in the same direction[11]. Although more

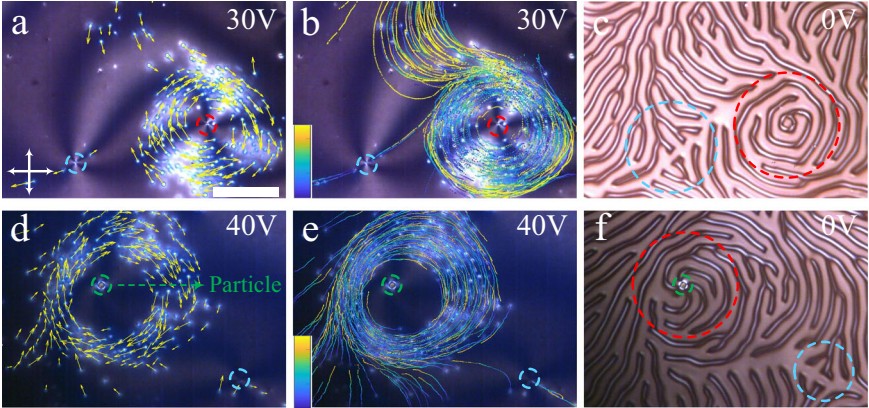

**Fig. 10 Circular motion of directron flocks.** Polarizing micrographs of circulating motion of directrons around (**a**) a $s = +1$ defect at $f = 100$ Hz, $U = 30$ V and (**d**) a micro-particle at $f = 60$ Hz, $U = 40$ V with their velocities represented as yellow arrows. Polarizing micrographs of circulating motion of directrons around (**b**) a $s = +1$ defect and (**e**) a micro-particle with their trajectories colored with time corresponding to the color bar. The color bars change linearly from (**b**) $t = 0$ s (dark blue) to $t = 10$ s (light yellow) and (**e**) $t = 0$ s (dark blue) to $t = 8.6$ s (light yellow), respectively. **c**, **f** Polarizing micrographs of the fingerprint textures at $U = 0$ V. Scale bar 100 µm. The crossed polarizers are indicated as white arrows. The $s = +1$ defects and $s = -1$ defects are circled with red and blue dashed lines, respectively. The micro-particle is circled with a green dashed line.

complicated collective behaviors of the skyrmions were realized by the authors later through optical manipulation[66], the setup of the experimental system is relatively complicated, which includes optical manipulations tools, such as laser tweezers, and photosensitive chiral dopants. Moreover, the types of collective behaviors in those systems are still limited where self-organization of flocks, fission and fusion process, density dependent collective motion, swirling and vortices, etc., are not reported.

One may suspect that the directrons observed here are "torons" or "skyrmions"[23]. However, the solitons here show a frequency-dependent stability, i.e., the solitons disappear at fixed voltages by changing the frequency only. The voltage threshold of the generation of the solitons is also dependent on frequency (Fig. 1c and Supplementary Fig. 8). Such a frequency-dependent stability is not observed if the solitons are torons or skyrmions since their stability is guaranteed by their topological structure and is independent on the frequency of the applied electric field[28]. Moreover, the generation of torons or skyrmions usually requires a symmetry-breaking transition of the LC system, which can be induced by phase transition[23], Freedericksz transition[28], or strong electro-hydrodynamic instabilities[11]. However, since the dielectric anisotropy of the LC media here is negative, there is no Freedericksz transition in cells of planar alignment. For samples of homeotropic alignment, the solitons have a threshold much higher than the threshold of the Freedericksz transition (Supplementary Fig. 8), i.e., before the formation of the solitons, the LC sample has changed from the homeotropic structure to the translationally invariant configuration. Due to the conservation of the topological charge of the LC system, one cannot continuously create topological solitons from a topologically trivial homogeneous state (no electro-hydrodynamic instability is observed during the formation of solitons). Furthermore, torons or skyrmions can only exist at relatively low voltages. If the applied voltage is too high, they will either disappear (in cells of planar alignment)[28] or transform into pairs of umbilic defects (in cells of homeotropic alignment)[67]. However, the solitons here are stable even at very large voltages ($U > 60$ V at $f = 100$ Hz). Thus, one can conclude that the solitons here are not torons or skyrmions, but directrons.

Topological trivial particle-like localized dissipative solitary director waves, i.e., directrons, have received great attention recently due to their intriguing nonlinear dynamic properties and potential applications in various areas such as microfluidics and optics[29–34]. We have shown previously that by adding chirality to the nematic LC system, the directrons exhibit a wave-particle duality that they can either pass through each other like waves or collide like particles[31,32]. Here, we show that by further decreasing the pitch of the LC system, the directrons can behave like true particles which collide without passing through each other and exhibit short-range repulsive but long-range attractive interactions. Unlike the directrons in achiral nematics and chiral nematics of large pitches reported earlier, which can only move in specific directions[29–31], the directrons here are equally likely to move in any direction in the $xy$ plane and can easily change their directions during motion due to the small pitch chiral system, thus leading to the emergence of various collective behaviors. The directrons are propelled by the periodic oscillation of the director field, and their moving direction is determined by their symmetry-breaking structure. The directrons self-organize into finite flocks which move in random directions with no correlations. They collide and fragment during motion. Within flocks, individual directrons move coherently in the same direction. At high packing fractions, directrons start from random motion and synchronize their motion and develop polar ordering through many-body interactions. The traveling directrons even self-organize into large-scale transient vortices and jets. The motion of the directrons cause large fluctuations in population density, which exhibit an anomalous scaling with system sizes. Impressively, the collective motion of the directrons can be facilely controlled by the electric field and topological defects. Our findings show that active matter composed of directrons can be used as an excellent model system for studying general principles of collective motion. The facile fabrication, control, and observation of our system also make it potentially promising for studying many other non-equilibrium phenomena, such as non-equilibrium phase transition[68], motility induced phase separation[69], out-of-equilibrium self-organization[70], etc. Since the directrons are actually self-localized director deformations and can be easily controlled through electric fields, they may even lead to new applications in microfluidics, such as micro-cargo transport[31,71], and electro-optics, such as tunable holographic optical devices.

## Methods

**Materials**. The chiral nematic is obtained by mixing a nematic liquid crystal ZLI-2806 (Merck) and a chiral dopant ZLI-811 (Xianhua, China). The nematic ZLI-

2806 shows a phase sequence on cooling of Isotropic (100 °C) Nematic (−20 °C) Crystal[63]. The components of the dielectric permittivity and conductivity of ZLI-2806 at $f = 4$ kHz and room temperature are $\varepsilon_{\parallel} = 3.0$, $\varepsilon_{\perp} = 7.6$, $\sigma_{\parallel} = 1.9 \times 10^{-8}\ \Omega^{-1} m^{-1}$, $\sigma_{\perp} = 6.0 \times 10^{-9}\ \Omega^{-1} m^{-1}$, respectively[31]. The pitch of the LC mixture is calculated according to the equation $p = 1/(HTP \times c)$, where $c$ is the weight concentration of the chiral dopant, and $HTP = -8.3\ \mu m^{-1}$ represents the helical twisting power of the chiral dopant[31]. The weight concentration of chiral dopant is ~6% in chiral nematic mixture of $p \sim 2\ \mu m$ and ~1.2% in chiral nematic mixture of $p \sim 10\ \mu m$. The experimental cells (AWAT, Poland) are prepared with a planar alignment layer. The inner surfaces of the cells are covered with polyimide and rubbed in a specific direction. The cell gaps are measured by the thin-film interference method[72]. In the experiments of circular motion of directrons induced by umbilic defects, the pitch of the LC mixture $p \sim 10\ \mu m$ and the cells (North LCD) used are prepared with homeotropic alignment with a cell gap $d \sim 9.4\ \mu m$.

**Generation of directrons**. The LC mixture is heated to the isotropic phase and filled into cells by capillary action. The filled sample is kept at 80 °C on a hot stage (LTSE350, Linkam) controlled by a temperature controller (TP 94, Linkam). The directrons are generated by applying an AC electric field directly to the LC cell using a waveform generator (33220 A, Agilent) and a home-built amplifier.

**Brownian motion of colloidal micro-particles**. A small amount of micro-particles with a homogeneously distributed diameter (~3 μm) are dispersed in the chiral nematic media ($p = 2\ \mu m$). The sample is kept at 75 °C and the motion of the micro-particles is tracked by the camera for ~2 min. The frame rate of the camera is tuned to 150 frames per second, which gives, overall, ~18,000 trajectory steps for measurement. The mean square displacement $(\Delta r^2(\tau))$ of the micro-particle is calculated, which grows linearly with time lag $\tau$ as $\Delta r^2(\tau) = 6L\tau$, where $L$ is the diffusion coefficient.

**Microscopic observations**. All images and movies are captured through polarizing transmission-mode optical microscopy using a Leica OPTIPOL microscope equipped with a charge-coupled device camera (UI-3360CP-C-HQ, uEye Gigabit Ethernet).

**Measurement of fluid flows**. To detect the potential fluid flows, small amounts of a fluorescent dye (0.04 wt%) are doped in the LC system. However, as we mentioned before, the formation of the directrons is closely related to the conductivity of the LC material and the fluorescent dye usually greatly changes the conductivity of LCs. As a result, no directron is observed; instead, a global 2D grid convective pattern is formed in the doped system.

We then doped quantum dots (0.005 wt%) into the LC media. The voltage thresholds of the directrons are greatly decreased by the doping, and once the voltage is >20 V, electro-convection patterns emerge and fill up the whole sample. The sample is then characterized by fluorescent microscope; however, nothing is observed. We then increased the concentration of quantum dots to 0.1 wt%. At such a high concentration, no directrons are generated and the sample is exhibiting electro-convection patterns throughout. Although the concentration is too high to induce the directrons, nothing is observed under the fluorescent microscope except a very weak blueish background. This may be because the quantum dots are too small to be seen.

We further doped micro-particles (diameter 3 μm) into the LC system. The doping slightly decreases the voltage threshold of the directrons. We find that individual micro-particles are very easily pinned on the glass substrates of the cell by applying the electric field. On the other hand, some aggregates of micro-particles and dusts move at relatively high voltages. However, it is always observed that there are directrons attached on those moving particles. The directrons always firstly nucleated at particles or dusts and then move them. Such a behavior has also been reported by Li et al.[71] and is called "soliton-induced liquid crystal enabled electrophoresis". So, it is not clear whether the particles are moved by the directrons or some fluid flows, or combinations thereof. However, no obvious motion of particles is observed before the emergence of directrons. The moving aggregates also stop moving once the directrons disappear by slightly decreasing the applied voltage. So one can conclude that at least before the formation of the directrons, there is no fluid flow present.

**Data analysis**. The movies are analyzed for positions of directrons using open-source software ImageJ-FIJI and its plugin "TrackMate". The position data is then processed with ORIGIN and MATLAB to characterize the dynamics of the directrons. The velocity order parameter is defined as $S_v = \left| \sum_j^N \mathbf{v}_j \right| / (N \times v_s)$, where $N$ is the total number of directrons in the field of view and $v_s$ is the absolute value of the velocity of coherently moving directrons[28]. The spatial velocity correlation function is defined as $g_v(r) = \left\langle \sum_{i,j}^N \left( \mathbf{v}_i \times \mathbf{v}_j \right) \delta \left[ r - r_{ij} \right] \right\rangle / \left\langle \sum_i^N \mathbf{v}_i \times \mathbf{v}_i \right\rangle$[9]. The temporal velocity correlation function is defined as $C_v(\triangle t) = \left\langle \sum_i^N \mathbf{v}_i(t) \times \mathbf{v}_i(t + \triangle t) \right\rangle / \left\langle \sum_i^N \mathbf{v}_i(t) \times \mathbf{v}_i(t) \right\rangle$. The giant number fluctuations and the scaling trend for each voltage in Fig. 8a are obtained by analyzing the directron number density versus time for 30 areas of different sizes, ranging from 53 μm × 53 μm to

367 μm × 367 μm. The time period over which the fluctuations are characterized for each data point in Fig. 8a is 500 s[28]. The hexatic bond orientational correlation function is defined as $g_6(r) = \left\langle \sum_{i,j}^N \psi_6^*(\mathbf{r}_i) \psi_6(\mathbf{r}_j) \delta(r - r_{ij}) \right\rangle$[73], where $\psi_6(\mathbf{r}_j) = \left( \frac{1}{m_j} \right) \sum_{k=1}^{m_j} e^{i6\theta_{jk}}$ is the local hexatic bond orientational order parameter. $m_j$ runs over the neighbors of directron $j$ (defined as being closer than a threshold distance), and $\theta_{jk}$ is the angle between the $j$-$k$ bond and an arbitrary axis. The radial distribution function is defined as $g_v(r) = \frac{S}{2\pi r N^2} \left\langle \sum_{i,j,i \neq j}^N \delta \left( r - r_{ij} \right) \right\rangle$, where $S$ is the area[9]. The pair interaction potential function ($U(r)$) is evaluated from the Boltzmann distribution $U(r) - k_B T \ln[g(r)]$[11,74]. We first measured the radial distribution function ($g(r)$) at very low density (low applied voltages) and then calculated the pair interaction function through the Boltzmann distribution. The critical frequency $f_c = \sqrt{\xi^2 - 1}/\tau_M$, where $\tau_M = \varepsilon_0 \varepsilon/\sigma$ is the Maxwell relaxation time for planar cells, $\varepsilon_0 = 8.85 * 10^{-12}$ Fm$^{-1}$. $\xi^2 = \left( 1 - \frac{\sigma}{\sigma_{\parallel}} \frac{\varepsilon_{\parallel}}{\varepsilon} \right) \left( 1 + \frac{\alpha_2}{\eta_c} \frac{\varepsilon_{\parallel}}{\Delta \varepsilon} \right)$ is the material parameter that depends on conductivities ($\sigma_{\perp}$ and $\sigma_{\parallel}$), permittivities ($\varepsilon_{\perp}$ and $\varepsilon_{\parallel}$), and viscous coefficients ($\alpha_2$ and $\eta_c$). Using the material data of ZLI-2806 measured at $f = 4$ kHz and room temperature, $1/\tau_M \sim 89$ Hz. The factor $\sqrt{\xi^2 - 1}$ is hard to determine exactly since both $\alpha_2$ and $\eta_c$ are not known. $\left( 1 - \frac{\sigma}{\sigma_{\parallel}} \frac{\varepsilon_{\parallel}}{\varepsilon} \right) \sim 0.875$ and $\frac{\varepsilon_{\parallel}}{\Delta \varepsilon} \sim -0.65$. We assume the ratio $-\frac{\alpha_2}{\eta_c}$ being on the order of 1 as Li et al. did in their work[30], the critical frequency $f_c \sim 128$ Hz.

## Data availability
The data generated in this study have been deposited in the figshare database in the following URL: https://figshare.com/articles/dataset/Electrically_tunable_collective_motion_of_dissipative_solitons_in_chiral_nematic_films/19213083.

## Code availability
All custom codes used for data processing have been deposited in the public repository Zenodo and are available from the following URL: https://zenodo.org/record/6361737#.YjGtWI_P2M8.

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

## Acknowledgements

Y. S. would like to acknowledge the China Scholarship Council for support (201806310129). We thank professor Jianren Lu and Dr. Xuzhi Hu for helping us with the fluorescent microscopy. We also thank professor Mark Dickinson for helping us with laser tweezers.

## Author contributions

Y. S. conceived and carried out the experimental investigations, analyzed the experimental results, and wrote the manuscript. I. D. supervised the investigations and contributed through discussions and writing the manuscript.

## Competing interests

The authors declare no competing interests.

**Additional information**

