## [Peer Review File · Nature Communications]

REVIEWER COMMENTS

Reviewer #1 (Remarks to the Author):

Out-of-equilibrium collective behavior is one of the hottest research areas. This article contributes analysis of solitons in liquid crystals from such a standpoint of view. At this moment I cannot recommend it for publication in this or any other journal, but I hope the manuscript can be improved. My biggest concern is that this and a series of other works on so-called "dissipative solitons" report on zoology of behaviors without a solid understanding of nature of these "dissipative solitons". A recent work by Pikin (<https://link.springer.com/article/10.1134/S1063776121040257> , JETP 132, 637–640 (2021)) states that there are no "theoretical explanations" and that the mechanism of formation is not clear. I completely agree with this assessment. The studies so far, by these authors and others, are done at a level of filming birefringent features moving around, but what are they and why they exist if they have solitonic nature? Authors have a chiral nematic version(s) of "dissipative solitons", but it appears to be even more puzzling what they are and why they appear; however, as I will argue below, they cannot be the same thing in planar and homeotropic confinement (though authors seem to imply this). Authors use some commercial glass cells, where they are restricted in what can be seen under a microscope, not even using high NA objectives. Unfortunately, they do not make an effort of convincingly figuring out the nature and structure of these solitons.

There are two types of cells. For homeotropic glass cells, from everything I see authors describing, I have no doubt that these generated solitons are torons or perhaps skyrmions (if the anchoring on substrates would be very weak) - the topological solitons as authors call them. I do not agree with the authors' way of distinguishing the "dissipative" and "topological" as the stability of topological solitons is only possible in some parameter range, including presence of external fields. What authors have might be topological solitons too, and I think they are torons. Now, seeing how authors interpret "torons" as "dissipative solitons" for which they draw (questionable) director structure in their previous paper for planar cells, I am very concerned. I strongly urge them to re-visit the structure of the solitons in planar cholesteric cells as well. Furthermore, I disagree with the model authors draw in ref. 46. I recommend that authors prepare samples of different thickness and pitch, do different types of imaging (including confocal) and numerical modeling, if possible, to clearly figure out the structure and the reason for motions under applied fields.

Is there fluid motion or just rotational director dynamics in the experiment? Did authors use tracer particles to detect flows and how these flows correlate or not with motion directions of single or many solitons? I note that a recent theory by Selinger shows how flows are not necessary for soliton motions: <https://arxiv.org/abs/2109.07314> - confirming this would be both timely and important, but of course this could be different for what authors study. If authors have some flows - is it just flow carrying solitons, with material transport involved in the forms of flows, and the solitons along with that? If so, is it still interesting? Certainly, this might not be as emergent and relevant to active matter in this case as all authors would probe would be these fluid flow currents and how smaller micro-rivers combine into bigger ones, not the collective self-propulsions. Authors should explore how the flows and mass transport, if existent, correlate with soliton and collective soliton motions.

Page 1, 1st paragraph – saying that "so far great achievement... has been achieved through numerical modeling..." – this is not accurate now (statements like this were appropriate 10 years ago) as active matter is the most active branch of the soft side of condensed matter now and hundreds of outstanding experiments and analytical models are reported every

month. Authors should refer to many elegant experimental and theory works by Dogic, Fraden, Irvine, Vitelli, Yeomans, Bowick...

Authors refer to their solitonic structures as bullets, but the original historic name for these solitons was "butterfly", given by Cladis. I have seen some other names, like "directron". Perhaps there could be a good reason to give a different name once the structure is known, but so far the "baterfly" name seemed to be consistent with some images in nematics. The work by Cladis and her team [Phys. Lett. A 235, 508–514 (1997).] should be referenced as the first experimental study of such solitons. I note that authors were kind of critical and looking down on this work (and this style I would discourage) in their Comp Phys article, saying "It should be noted that a similar phenomenon was earlier observed by Brand et al.³², who reported localized formations in the shape of "butterflies" that could move in the plane of the cell. However, neither the director structure ... was revealed in their report." I must say that, unfortunately, this structure in both planar and homeotropic cells that authors study is still unknown as well, though I think the solitons in homeotropic cells are torons. 30 years after Cladis' work, authors have more tools to uncover the structures of solitons they study.

Authors say "Different kinds of solitons have been produced in LCs. However, most of them are immobile" - I would disagree as I have seen books & hundreds of articles on mobile liquid crystal solitons, many more than what was devoted to static solitons.

Authors study all kinds of collective behaviors and refer to herds and swarms and similar formations of particles that only interact through bumping into each other, like in some of the original active matter toy models. However their medium is a liquid crystal, the soliton structure in director field implies elastic energy costs - here must be elastic interactions between them - what is their role? Recent work from LosAlamos even predicts spin ice formation in solitons due to such interactions: [10.1103/PhysRevLett.126.047801](https://doi.org/10.1103/PhysRevLett.126.047801) , DOI: [10.1039/b000000x](https://doi.org/10.1039/b000000x) If there are flows, are there hydrodynamic interactions as well? What is the Ericksen number? I urge authors to probe what is the nature of interactions between solitons? Elastic, hydrodynamic, electrostatic?

Authors refer to the texture in homeotropic cell as corresponding to umbilics when in-plane component of field is induced in hom. cells. However, it could be also that authors create the so-called dowser texture (see many elegant works by Pieransky & others): <https://onlinelibrary.wiley.com/doi/abs/10.1002/9781119850809.ch4> At least the possibility should be discussed and authors should say how they checked. The umbilic in this case could be a hedgehog point defect, which might be explaining the fact that it becomes a generation site for the solitons seen in one of the movies...

Page 2, top paragraph – authors motivate their study by noting that the prior work on soliton dynamics had a faster synchronization time scale, but they prefer longer than seconds. Indeed, movies in the present article seem to be 10X sped up, so the system is slow indeed. However, it would seem that this new study would be an incremental accomplishment if this was the only progress made as compared to that old NatCom paper. Moreover, the same author, H. Sohn, and other authors have done much more in the field of active/driven solitons (PNAS 117, 6437 (2020), Opt Express 28, 6306 (2020), Opt. Exp. 27, 29055 (2019), PR E 97, 052701 (2018)...), so authors should do a more thorough review of this literature and better describe the novelty of what they do to merit publication in NatCom. Moreover, some of these other papers had regimes even closer to what authors study here, including slower synchronization. The present study indeed seems to be VERY inspired by the works of Sohn as it reports similar types of characterizations that Sohn did in his series of papers, as well as characterization of the same effects (velocity order parameters, giant number fluctuations as in Nat Comm, motions around induced umbilical defects in the OptExp. Article, hexatic like ordering in PNAS...). On top of this, as I already

mentioned, I believe some of the solitons authors study are the same torons that Sohn studied in cells of similar thickness, pitch and same material ZLI-2806... It does not mean authors have no novel results – they clearly see something different too, which should be highlighted and put on shelves distinguishing prior art from what is done in this work.

In summary, I cannot recommend the manuscript in its present form, but I will be happy to look at responses and revisions.

Reviewer #2 (Remarks to the Author):

The manuscript demonstrates rich collective behavior of 3D director solitons in a chiral nematic but the presentation makes it difficult to recommend publication.

- What are the noteworthy results?

A. Observation of random trajectories of solitons that arrange in coherent structures and measurement of giant density fluctuations are noteworthy results; however, lack of experimental characterization makes it difficult to understand the underlying mechanisms

- Will the work be of significance to the field and related fields? How does it compare to the established literature? If the work is not original, please provide relevant references.

B. The observations are new and interesting, but the presentation lacks depth on substance and mechanisms; comparison with other active systems are not always justified

- Does the work support the conclusions and claims, or is additional evidence needed?

C. Additional evidence is needed. The experimental results are described in a manner that makes it hard to understand the conclusions. For example, the manuscript does not even specify the concentration of the chiral dopant in the mixture. Furthermore, the anisotropy of dielectric permittivity and electric conductivity are important factors in electrohydrodynamics of liquid crystals in general and the formation of solitons in particular. However, the manuscript does not discuss the role of these anisotropies; the data on anisotropies are relegated to the end, making the main part of the text hard to understand. There is no discussion of how the length and width of the solitons affect the correlation functions and how the maxima and minima of these are related to the length and width of the solitons. It is not clear how the authors established that the director oscillates with the frequency of the applied field. It is not clear how the director distortions shown in Fig 2a were established.

- Are there any flaws in the data analysis, interpretation, and conclusions? - Do these prohibit the publication or require revision?

D. The presentation misses important points relevant to the system at hand (director field, its oscillations with the frequency of the applied field, the role of anisotropies, size of solitons, plausible mechanisms of interactions, etc.) and instead draws superficial links to other systems, such as flocks of birds.

- Is the methodology sound? Does the work meet the expected standards in your field?

E. It is not clear how the soliton trajectories are related to the local director on the

cholesteric pseudo-layers. It is not clear how the director field configuration was established. It is not clear how the oscillation frequency of the director was measured.

**- Is there enough detail provided in the methods for the work to be reproduced?
F. No, please see above.**

Other comments:

- 1. The Abstract must describe the system under study. The current text is too general to understand what the solitons are and which material they form in and under which circumstances. Terms such as "animal fluid" are ambiguous.**
- 2. The text relies heavily on the very general term "soliton". Although it is true that the observed objects are solitons, this general term is not descriptive enough of the findings, since there are many different solitons in Nature and laboratories. Why not use more specific terms?**
- 3. Statements such as "the director is uniformly self-assembled into a helical superstructure" are not scientifically sound. Self-assembled might be, for example, molecules or colloids, but the director is in a different category.**
- 4. What is the nature of interactions of solitons along the normal to the plates and in the plane of the cell?**
- 5. What is the origin of a "cutoff size" in the system?**
- 6. Why in some cases the increase of the field leads to more ordered structures and in other cases to chaotic behavior?**

Reviewer #3 (Remarks to the Author):

The manuscript entitled "Electrically tunable collective motion of dissipative solitons in chiral nematic films" by Yuan Shen and Ingo Dierking presents a collective motion of chiral nematic solitons. This is a kind of follow-up of the works on the nematic solitons reported by Li et al. and by the present authors. In this paper, the authors found that the motions of nematic solitons mimicking the collective behaviors established in many active matter systems, such as flocks of birds or schools of fishes. This concept works very well and may attract broad readership in the field of physics. especially at this timing that Prof. Parisi won this year's Nobel prize for physics. The experimental results in this work are well examined. Data processing is properly made and the consequence is scientifically sound. So basically I think this is a nice paper and should be published in Nature Communications. My only concern is, most of explanation in this paper is just qualitative consideration (only size distribution in Fig.5 is discussed well compared to others), and thus there is slight lack of quantitative physical analysis based on formulation. For instance, one of the intriguing points in the active matter physics is scale-free correlations like the flocks of birds. The authors have summarized the 2D correlation functions and their behavior on the relative coordinate r . So, the rescalability among the soliton flocks might be interesting to be discussed further in these data upon comparison with other scale-free systems.

More technically, I have several comments as follows;

- 1. If I understand correctly the experimental condition, the director field of the present system is twisting along the cell normal. Then, it is difficult to imagine how the solitons as schematized in Fig.1d can be distributed with twisting in the cell? It would be appreciated if the authors could improve this point by drawing the exact director distortion in a soliton in a better way (cross-sectional views may work, in my opinion).**
- 2. It would be also nice, if the initial director observation is described somewhere in the**

manuscript or in the supplementary information. The used cell was a planer cell with a homogeneous surface condition but no given preferable alignment direction. In such a case, I guess there are lots of line defects which may trap solitons (flocks).

Reviewer #4 (Remarks to the Author):

The authors present a number of interesting experiments, where large numbers of solitons in a chiral nematic liquid crystal are created by applying an external electric field of varying amplitude and frequency. The authors find that these solitons resemble flocks in active matter systems, because they experience collective motion, they are able to synchronize their motion and some eye-catching dynamic phenomena of schools of solitons are seen on the videos.

This work is without doubt very interesting and novel. This work is also timely, as active nematics are a topic of great current interest. In this sense this work is quite unique, because there are not many experimental settings that would display the behavior of active matter in liquid crystals. There is no doubt that this work opens a new direction in liquid crystal research, bridging the gap between the liquids and liquid crystal communities. The significance of this work is high.

While the topic presented is of great current interest, the manuscript itself has many deficiencies, which need to be removed before final decision on publication is reached. In particular I have the following questions and comments:

1. Why is the intriguing collective behavior of solitons observed in this experiment and not in many previous experiments on similar solitons by the same authors? What is the key difference here, in this experiment, compared to many other similar experiments? The authors have published several articles on solitons in chiral nematic liquid crystals, using practically the same LC materials and experimental cells. For example, the authors have studied ZLI-2806 and a chiral dopant ZLI-811 in previous experiments published in Comm.Phys. 2020, but have not reported schools of solitons. Is the reason in the chirality, i.e. the length of the pitch? This needs to be clarified to get broader insight into the emergence of active matter behavior of solitons.
2. On page #2 the authors give a brief and not convincing explanation of why the solitons move "Within the soliton, the director field oscillates with the frequency of the applied electric field due to the flexoelectric effect, breaking the mirror symmetry of the solitons and driving them to move through the uniform nematic bulk." This needs to be better explained, also the structure of a single soliton has to be clearly described, together with the role of dielectric anisotropy and conductivity anisotropy. These are mentioned in section Methods, but their role in soliton propulsion is not clarified.
3. The LC cell structure should be better explained and more accurate. The statement "The experimental setup is similar to that in LC displays, where a thin film of cholesteric LC is sandwiched between two pieces of glass substrates which is coated with an indium tin oxide (ITO) layer as electrodes and a rubbed polyimide layer as the planar alignment layer." is not clear, as the reader might not be familiar with what "planar alignment" is and how the helix is oriented with respect to the cell surface.
4. What is meaning of "large chirality" in paragraph starting with Dynamics of solitons? Is this the absolute value of the pitch? The authors indicate that omnidirectional movement of the solitons is due to "large chirality" but do not try to explain the reasons for such behavior.

5. In Figure 2(e) the authors claim that "Pair interaction potential function(extracted from the radial distribution function $g(r)$ shown in the inset) of the soliton...." How was the pair interaction calculated from $g(r)$? Please give full explanation of this important part of the manuscript.

6. Figure 5. needs more accurate description. Is it correct to understand that it represents a probability $P(n)$ that a flock of solitons will have n members? If so, please write it clearly, using full and understandable sentences.

7. The authors show circular motion of solitons in Figure 11, but it is not clearly stated that the liquid crystal is the same as for previous experiments, but has a slightly larger pitch. Flocks of solitons experiencing circular motion is observed, but it is not clear if these solitons are of the same sort as those studied in planar cells? If so, why do we see similar solitons at lower chirality, i.e. larger pitch, while the authors have claimed that flocks are observable for large chirality?

8. The authors mention that solitons could be used for "micro cargo transport". I would be convinced if the authors showed how solitons are able to move a microparticle in a LC.

The article is not well written, the structure is not clear, in some places the same issues are repeated several times. The narration of the article is poor. The article is descriptive with limited aim to explain the physics behind the observed phenomena.

REVIEWER COMMENTS

Reviewer #1 (Remarks to the Author):

Out-of-equilibrium collective behavior is one of the hottest research areas. This article contributes analysis of solitons in liquid crystals from such a standpoint of view. At this moment I cannot recommend it for publication in this or any other journal, but I hope the manuscript can be improved. My biggest concern is that this and a series of other works on so-called “dissipative solitons” report on zoology of behaviors without a solid understanding of nature of these “dissipative solitons”. A recent work by Pikin (<https://link.springer.com/article/10.1134/S1063776121040257> , JETP 132, 637–640 (2021)) states that there are no “theoretical explanations” and that the mechanism of formation is not clear. I completely agree with this assessment. The studies so far, by these authors and others, are done at a level of filming birefringent features moving around, but what are they and why they exist if they have solitonic nature? Authors have a chiral nematic version(s) of “dissipative solitons”, but it appears to be even more puzzling what they are and why they appear; however, as I will argue below, they cannot be the same thing in planar and homeotropic confinement (though authors seem to imply this). Authors use some commercial glass cells, where they are restricted in what can be seen under a microscope, not even using high NA objectives. Unfortunately, they do not make an effort of convincingly figuring out the nature and structure of these solitons.

There are two types of cells. For homeotropic glass cells, from everything I see authors describing, I have no doubt that these generated solitons are torons or perhaps skyrmions (if the anchoring on substrates would be very weak) - the topological solitons as authors call them. I do not agree with the authors’ way of distinguishing the “dissipative” and “topological” as the stability of topological solitons is only possible in some parameter range, including presence of external fields. What authors have might be topological solitons too, and I think they are torons. Now, seeing how authors interpret “torons” as “dissipative solitons” for which they draw (questionable) director structure in their previous paper for planar cells, I am very concerned. I strongly urge them to re-visit the structure of the solitons in planar cholesteric cells as well. Furthermore, I disagree with the model authors draw in ref. 46. I recommend that authors prepare samples of different thickness and pitch, do different types of imaging (including confocal) and numerical modeling, if possible, to clearly figure out the structure and the reason for motions under applied fields.

Is there fluid motion or just rotational director dynamics in the experiment? Did authors use tracer particles to detect flows and how these flows correlate or not with motion directions of single or many solitons? I note that a recent theory by Selinger shows how flows are not necessary for soliton motions: <https://arxiv.org/abs/2109.07314> - confirming this would be both timely and important, but of course this could be different for what authors study. If authors have some flows - is it just flow carrying solitons, with material transport involved in the forms of flows, and the solitons along with that? If so, is it still interesting? Certainly, this might not be as emergent and relevant to active matter in this case as all authors would probe would be these fluid flow currents and how smaller micro-rivers combine into bigger ones, not the collective self-propulsions. Authors should explore how the

flows and mass transport, if existent, correlate with soliton and collective soliton motions.

Page 1, 1st paragraph – saying that “so far great achievement... has been achieved through numerical modeling...” – this is not accurate now (statements like this were appropriate 10 years ago) as active matter is the most active branch of the soft side of condensed matter now and hundreds of outstanding experiments and analytical models are reported every month. Authors should refer to many elegant experimental and theory works by Dogic, Fraden, Irvine, Vitelli, Yeomans, Bowick...

Authors refer to their solitonic structures as bullets, but the original historic name for these solitons was “butterfly”, given by Cladis. I have seen some other names, like “directron”. Perhaps there could be a good reason to give a different name once the structure is known, but so far the “baterfly” name seemed to be consistent with some images in nematics. The work by Cladis and her team [Phys. Lett. A 235, 508–514 (1997).] should be referenced as the first experimental study of such solitons. I note that authors were kind of critical and looking down on this work (and this style I would discourage) in their Comp Phys article, saying “It should be noted that a similar phenomenon was earlier observed by Brand et al.³², who reported localized formations in the shape of “butterflies” that could move in the plane of the cell. However, neither the director structure ... was revealed in their report.” I must say that, unfortunately, this structure in both planar and homeotropic cells that authors study is still unknown as well, though I think the solitons in homeotropic cells are torons. 30 years after Cladis’ work, authors have more tools to uncover the structures of solitons they study.

Authors say “Different kinds of solitons have been produced in LCs. However, most of them are immobile” - I would disagree as I have seen books & hundreds of articles on mobile liquid crystal solitons, many more than what was devoted to static solitons.

Authors study all kinds of collective behaviors and refer to herds and swarms and similar formations of particles that only interact through bumping into each other, like in some of the original active matter toy models. However their medium is a liquid crystal, the soliton structure in director field implies elastic energy costs - here must be elastic interactions between them - what is their role? Recent work from LosAlamos even predicts spin ice formation in solitons due to such interactions: 10.1103/PhysRevLett.126.047801 , DOI: 10.1039/b000000x If there are flows, are there hydrodynamic interactions as well? What is the Ericksen number? I urge authors to probe what is the nature of interactions between solitons? Elastic, hydrodynamic, electrostatic?

Authors refer to the texture in homeotropic cell as corresponding to umbilics when in-plane component of field is induced in hom. cells. However, it could be also that authors create the so-called dowser texture (see many elegant works by Pieransky & others): <https://onlinelibrary.wiley.com/doi/abs/10.1002/9781119850809.ch4>

At least the possibility should be discussed and authors should say how they checked. The umbilic in this case could be a hedgehog point defect, which might be explaining the fact that it becomes a generation site for the solitons seen in one of the movies...

Page 2, top paragraph – authors motivate their study by noting that the prior work on soliton dynamics had a faster synchronization time scale, but they prefer longer than seconds. Indeed, movies in the present article seem to be 10X sped up, so the system is slow indeed. However, it would seem that this new study would be an incremental accomplishment if this was the only progress made as compared to that old NatCom paper. Moreover, the same author, H. Sohn, and other authors have done much more in the field of active/driven solitons (PNAS 117, 6437 (2020), Opt Express 28, 6306 (2020), Opt. Exp. 27, 29055 (2019), PR E 97, 052701 (2018)...), so authors should do a more thorough review of this literature and better describe the novelty of what they do to merit publication in NatCom. Moreover, some of these other papers had regimes even closer to what authors study here, including slower synchronization. The present study indeed seems to be VERY inspired by the works of Sohn as it

reports similar types of characterizations that Sohn did in his series of papers, as well as characterization of the same effects (velocity order parameters, giant number fluctuations as in Nat Comm, motions around induced umbilical defects in the OptExp. Article, hexatic like ordering in PNAS...). On top of this, as I already mentioned, I believe some of the solitons authors study are the same torons that Sohn studied in cells of similar thickness, pitch and same material ZLI-2806... It does not mean authors have no novel results – they clearly see something different too, which should be highlighted and put on shelves distinguishing prior art from what is done in this work.

In summary, I cannot recommend the manuscript in its present form, but I will be happy to look at responses and revisions.

Reviewer #2 (Remarks to the Author):

The manuscript demonstrates rich collective behavior of 3D director solitons in a chiral nematic but the presentation makes it difficult to recommend publication.

- What are the noteworthy results?

A. Observation of random trajectories of solitons that arrange in coherent structures and measurement of giant density fluctuations are noteworthy results; however, lack of experimental characterization makes it difficult to understand the underlying mechanisms

- Will the work be of significance to the field and related fields? How does it compare to the established literature? If the work is not original, please provide relevant references.

B. The observations are new and interesting, but the presentation lacks depth on substance and mechanisms; comparison with other active systems are not always justified

- Does the work support the conclusions and claims, or is additional evidence needed?

C. Additional evidence is needed. The experimental results are described in a manner that makes it

hard to understand the conclusions. For example, the manuscript does not even specify the concentration of the chiral dopant in the mixture. Furthermore, the anisotropy of dielectric permittivity and electric conductivity are important factors in electrohydrodynamics of liquid crystals in general and the formation of solitons in particular. However, the manuscript does not discuss the role of these anisotropies; the data on anisotropies are relegated to the end, making the main part of the text hard to understand. There is no discussion of how the length and width of the solitons affect the correlation functions and how the maxima and minima of these are related to the length and width of the solitons. It is not clear how the authors established that the director oscillates with the frequency of the applied field. It is not clear how the director distortions shown in Fig 2a were established.

- Are there any flaws in the data analysis, interpretation, and conclusions? - Do these prohibit the publication or require revision?

D. The presentation misses important points relevant to the system at hand (director field, its oscillations with the frequency of the applied field, the role of anisotropies, size of solitons, plausible mechanisms of interactions, etc.) and instead draws superficial links to other systems, such as flocks of birds.

- Is the methodology sound? Does the work meet the expected standards in your field?

E. It is not clear how the soliton trajectories are related to the local director on the cholesteric pseudo-layers. It is not clear how the director field configuration was established. It is not clear how the oscillation frequency of the director was measured.

- Is there enough detail provided in the methods for the work to be reproduced?

F. No, please see above.

Other comments:

1. The Abstract must describe the system under study. The current text is too general to understand what the solitons are and which material they form in and under which circumstances. Terms such as “animal fluid” are ambiguous.

2. The text relies heavily on the very general term “soliton”. Although it is true that the observed objects are solitons, this general term is not descriptive enough of the findings, since there are many different solitons in Nature and laboratories. Why not use more specific terms?

3. Statements such as “the director is uniformly self-assembled into a helical superstructure” are not scientifically sound. Self-assembled might be, for example, molecules or colloids, but the director is in a different category.

4. What is the nature of interactions of solitons along the normal to the plates and in the plane of the cell?

5. What is the origin of a “cutoff size” in the system?

6. Why in some cases the increase of the field leads to more ordered structures and in other cases to

chaotic behavior?

Reviewer #3 (Remarks to the Author):

The manuscript entitled "Electrically tunable collective motion of dissipative solitons in chiral nematic films" by Yuan Shen and Ingo Dierking presents a collective motion of chiral nematic solitons. This is a kind of follow-up of the works on the nematic solitons reported by Li et al. and by the present authors. In this paper, the authors found that the motions of nematic solitons mimicking the collective behaviors established in many active matter systems, such as flocks of birds or schools of fishes. This concept works very well and may attract broad readership in the field of physics, especially at this timing that Prof. Parisi won this year's Nobel prize for physics. The experimental results in this work are well examined. Data processing is properly made and the consequence is scientifically sound. So basically I think this is a nice paper and should be published in Nature Communications. My only concern is, most of explanation in this paper is just qualitative consideration (only size distribution in Fig.5 is discussed well compared to others), and thus there is slight lack of quantitative physical analysis based on formulation. For instance, one of the intriguing points in the active matter physics is scale-free correlations like the flocks of birds. The authors have summarized the 2D correlation functions and their behavior on the relative coordinate r . So, the rescalability among the soliton flocks might be interesting to be discussed further in these data upon comparison with other scale-free systems.

More technically, I have several comments as follows;

1. If I understand correctly the experimental condition, the director field of the present system is twisting along the cell normal. Then, it is difficult to imagine how the solitons as schematized in Fig.1d can be distributed with twisting in the cell? It would be appreciated if the authors could improve this point by drawing the exact director distortion in a soliton in a better way (cross-sectional views may work, in my opinion).
2. It would be also nice, if the initial director observation is described somewhere in the manuscript or in the supplementary information. The used cell was a planer cell with a homogeneous surface condition but no given preferable alignment direction. In such a case, I guess there are lots of line defects which may trap solitons (flocks).

Reviewer #4 (Remarks to the Author):

The authors present a number of interesting experiments, where large numbers of solitons in a chiral nematic liquid crystal are created by applying an external electric field of varying amplitude and frequency. The authors find that these solitons resemble flocks in active matter systems, because they experience collective motion, they are able to synchronize their motion and some eye-catching dynamic phenomena of schools of solitons are seen on the videos.

This work is without doubt very interesting and novel. This work is also timely, as active nematics are a topic of great current interest. In this sense this work is quite unique, because there are not many experimental settings that would display the behavior of active matter in liquid crystals. There is no doubt that this work opens a new direction in liquid crystal research, bridging the gap between the liquids and liquid crystal communities. The significance of this work is high.

While the topic presented is of great current interest, the manuscript itself has many deficiencies, which need to be removed before final decision on publication is reached. In particular I have the following questions and comments:

1. Why is the intriguing collective behavior of solitons observed in this experiment and not in many previous experiments on similar solitons by the same authors? What is the key difference here, in this experiment, compared to many other similar experiments? The authors have published several articles on solitons in chiral nematic liquid crystals, using practically the same LC materials and experimental cells. For example, the authors have studied ZLI-2806 and a chiral dopant ZLI-811 in previous experiments published in *Comm.Phys.* 2020, but have not reported schools of solitons. Is the reason in the chirality, i.e. the length of the pitch? This needs to be clarified to get broader insight into the emergence of active matter behavior of solitons.

2. On page #2 the authors give a brief and not convincing explanation of why the solitons move "Within the soliton, the director field oscillates with the frequency of the applied electric field due to the flexoelectric effect, breaking the mirror symmetry of the solitons and driving them to move through the uniform nematic bulk." This needs to be better explained, also the structure of a single soliton has to be clearly described, together with the role of dielectric anisotropy and conductivity anisotropy. These are mentioned in section Methods, but their role in soliton propulsion is not clarified.

3. The LC cell structure should be better explained and more accurate. The statement "The experimental setup is similar to that in LC displays, where a thin film of cholesteric LC is sandwiched between two pieces of glass substrates which is coated with an indium tin oxide (ITO) layer as electrodes and a rubbed polyimide layer as the planar alignment layer." is not clear, as the reader might not be familiar with what "planar alignment" is and how the helix is oriented with respect to the cell surface.

4. What is meaning of "large chirality" in paragraph starting with Dynamics of solitons? Is this the absolute value of the pitch? The authors indicate that omnidirectional movement of the solitons is due to "large chirality" but do not try to explain the reasons for such behavior.

5. In Figure 2(e) the authors claim that "Pair interaction potential function(extracted from the radial distribution function $g(r)$ shown in the inset) of the soliton...." How was the pair interaction calculated from $g(r)$? Please give full explanation of this important part of the manuscript.

6. Figure 5. needs more accurate description. Is it correct to understand that it represents a probability $P(n)$ that a flock of solitons will have n members? If so, please write it clearly, using full

and understandable sentences.

7. The authors show circular motion of solitons in Figure 11, but it is not clearly stated that the liquid crystal is the same as for previous experiments, but has a slightly larger pitch. Flocks of solitons experiencing circular motion is observed, but it is not clear if these solitons are of the same sort as those studied in planar cells? If so, why do we see similar solitons at lower chirality, i.e. larger pitch, while the authors have claimed that flocks are observable for large chirality?

8. The authors mention that solitons could be used for "micro cargo transport". I would be convinced if the authors showed how solitons are able to move a microparticle in a LC.

The article is not well written, the structure is not clear, in some places the same issues are repeated several times. The narration of the article is poor. The article is descriptive with limited aim to explain the physics behind the observed phenomena.

We thank you and the reviewers for a thorough evaluation of our work. We are grateful to the valuable comments which help us to improve the present work. Below we list all the comments and our replies. All the changes throughout the manuscript are marked with track changes.

Reviewer #1 (Remarks to the Author):

Q 1.1 out-of-equilibrium collective behavior is one of the hottest research areas. This article contributes analysis of solitons in liquid crystals from such a standpoint of view. At this moment I cannot recommend it for publication in this or any other journal, but I hope the manuscript can be improved. My biggest concern is that this and a series of other works on so-called “dissipative solitons” report on zoology of behaviors without a solid understanding of nature of these “dissipative solitons”. A recent work by Pikin (<https://link.springer.com/article/10.1134/S1063776121040257> , JETP 132, 637–640 (2021)) states that there are no “theoretical explanations” and that the mechanism of formation is not clear. I completely agree with this assessment. The studies so far, by these authors and others, are done at a level of filming birefringent features moving around, but what are they and why they exist if they have solitonic nature? Authors have a chiral nematic version(s) of “dissipative solitons”, but it appears to be even more puzzling what they are and why they appear; however, as I will argue below, they cannot be the same thing in planar and homeotropic confinement (though authors seem to imply this). Authors use some commercial glass cells, where they are restricted in what can be seen under a microscope, not even using high NA objectives. Unfortunately, they do not make an effort of convincingly figuring out the nature and structure of these solitons.

A 1.1 Firstly, we would like to thank the reviewer for spending his/her precious time on evaluating our work. We also thank the reviewer for his/her comments and giving us the opportunity to improve our manuscript. We agree with the reviewer’s and Pikin’s opinion that “the underlying mechanism of the formation of the ‘dissipative solitons’ is still not very clear and requires further experimental and theoretical investigations.” According to Pikin’s opinion, the generation of the solitons may be due to the interaction of injected electron clouds with nematic molecules (JETP 132, 637–640 (2021)). However, the charge injection usually happens under a DC electric field or AC electric field with low frequency (M. Nakagawa, J. Phys. Sco. Jpn, 52, 3773-3781, 1983; 52, 3782-3789, 1983), which is suppressed as soon as the oscillation frequency of the applied field exceeds a few cycles (F. Rondelez, Thesis, University de Paris, Orsay, 1970). According to the experiments reported by us and other groups previously (Refs [44-50]), the solitons can be induced at relatively high frequencies (up to 800 Hz or even higher). But the possibility of charge injection cannot be excluded since it may still happen at relatively high voltages. To totally exclude the possibility of charge injection, further experiments are required, for instance, one can make a cell with “blocking” electrodes (e.g. covering the electrodes with thin layers of pyrex glass) (H. Arnould-netillard and F. Rondelez, Mol. Cryst. Lid. Cryst., 26, 11-31, 1974). On the other hand, according to our and other groups’ previous works (Refs [44-50]), the solitons are nonsingular self-localized director perturbations, i.e., within the solitons, the director is deformed from the homogeneous state and oscillates with the frequency of the applied AC field and their formation may be attributed to the

effect of flexoelectric polarization and the accumulation of space charges. In the present work, the LC media used is ZLI-2806, which belongs to the (-,+) type, which means that the dielectric anisotropy is negative but the conductivity anisotropy is positive, i.e., $\Delta\varepsilon = \varepsilon_{\parallel} - \varepsilon_{\perp} < 0$ and $\Delta\sigma = \sigma_{\parallel} - \sigma_{\perp} > 0$, respectively. According to the Carr-Helfrich mechanism, conventional electro-convective patterns are usually generated in such systems. However, the solitons are generated instead in our case. We think this is due to the low conductivity anisotropy of our LC media ($\sim 1.3 \times 10^{-8} \Omega^{-1}\text{m}^{-1}$), which is relatively small compared to the ones used in studies of electro-convection ($\sim 10^{-7} \Omega^{-1}\text{m}^{-1}$) (refs [57,58]). According to ref [49], the solitons only occur in the limited range of moderate conductivity ($0.8 \times 10^{-8} < \sigma < 4 \times 10^{-8} \Omega^{-1}\text{m}^{-1}$). If the conductivity is higher than this range, only global electro-convective patterns are observed. The significance of this small conductivity anisotropy can be understood by considering the coupling between the electric field and the space charges. According to the Carr - Helfrich electro-convection mechanism, the positive conductivity anisotropy and the bend fluctuation in a nematic induce ion segregation and forms space charges which are high and uniformly distributed in space. These space charges produce transverse Coulomb forces which offset the normal elastic and dielectric torques and cause instability, usually in the form of space-filling periodic stripes (electro-convection pattern). However, due to the relatively low conductivity of our samples, there is not sufficient charge accumulation to produce a strong enough dielectric torque to induce a global electro-convection pattern. Instead, the director field around the space charges is locally deformed, inducing the flexoelectric polarization. As a result, the director within the local deformations oscillates with the frequency of the applied electric field (refs [44-46]), leading to the formation of the solitons.

As suggested by the reviewer, we have characterized the solitons with a higher NA objective. Although the POM images of the solitons characterized by this objective are not very clear because the thick glass substrates of the LC cell make focusing difficult and the light intensity of our microscope is not strong enough. We also want to do more characterizations of the soliton structure through different equipment such as confocal microscopy. However, the only available equipment to characterize the structure of the solitons in our lab is the polarizing microscope. So based on these POM images of the solitons and the formation mechanism of the solitons given above, we have deduced the director structure of the solitons (Supplementary Fig. 2). We have also added a discussion to demonstrate the formation of the solitons and their structures (in the first and second parts of the results section).

Q 1.2 There are two types of cells. For homeotropic glass cells, from everything I see authors describing, I have no doubt that these generated solitons are torons or perhaps skyrmions (if the anchoring on substrates would be very weak) - the topological solitons as authors call them. I do not agree with the authors' way of distinguishing the "dissipative" and "topological" as the stability of topological solitons is only possible in some parameter range, including presence of external fields. What authors have might be topological solitons too, and I think they are torons. Now, seeing how authors interpret "torons" as "dissipative solitons" for which they draw (questionable) director structure in their previous paper for planar cells, I am very concerned. I strongly urge them to re-visit the structure of the solitons in planar cholesteric cells as well. Furthermore, I disagree with the model authors draw in ref. 46. I recommend that authors prepare samples of different thickness and pitch, do different types of imaging (including confocal) and numerical modeling, if possible, to clearly figure out the structure and the reason for motions under

applied fields.

A1.2 We thank the reviewer for his/her comments and suggestions. The solitons here may look like torons or skyrmions to some extent, however, we doubt that the solitons in either planar cells or homeotropic cells are torons. Here are our reasons. As far as we are aware, the topological solitons, such as torons, are local frustration or twist of the helical director field of chiral nematics, in which the director field cannot be continuously deformed to the uniform state. They can usually be generated by heating the chiral nematic into isotropic phase and then cool it down to the nematic phase, or by inducing strong electro-hydrodynamic instabilities in the chiral nematic. Both methods require an abrupt breaking of the symmetry of the nematic phase. This is due to the conservation of the topological charge of the liquid crystal (LC) system, i.e., one cannot continuously create topological solitons from a topologically trivial uniform state. However, in our experiment, for the samples with homogeneous alignment, because the dielectric anisotropy of the LC media is negative, applying an electric field parallel to the helical axis can only stabilize the planar texture and cannot induce such a symmetry breaking (no typical electro-hydrodynamic instability is observed except the solitons). The helical axis of the chiral nematic stays perpendicular to the cell substrates, and thus cannot induce any topological twist or frustration of the director field. On the other hand, for the samples with homeotropic alignment, although the symmetry of the sample is broken due to the Freedericksz transition, the solitons in our case do not emerge right after the transition. Also the threshold of the Freedericksz transition (2.2V at $f=100\text{Hz}$) is much smaller than the threshold of the generation of the solitons (20.0 V at $f=100\text{Hz}$) (Figure below). This means that before the formation of the solitons, the sample is actually in the translationally invariant configuration, which is then basically similar to the samples with homogeneous alignment. Furthermore, the torons in refs [25, 42] are driven by relatively low voltages ($\sim 4.0\text{ V}$, cell gap $\sim 10\ \mu\text{m}$). Actually, the torons can only exist at relatively low voltages. If the dielectric anisotropy of the LC host is negative, for increasing the applied voltage, the torons will be continuously distorted in homeotropic cells and eventually turn into a pair of umbilic defects (T. Nagaya, et al., J. Phys. Soc. Jpn, 60, 1572-1578, 1991). On the other hand, if the dielectric anisotropy is positive, regardless of the anchoring condition of the LC cell (either planar or homeotropic), the torons will continuously shrink and eventually disappear because their topological structure is destroyed due to the unwinding of the helical structure induced by the large dielectric torque (ref [43]). For nematics with negative dielectric anisotropy confined in LC cells with homogeneous alignment, as far as we know, no skyrmions or torons has been reported so far because both the anchoring condition and external electric fields usually stabilize the planar texture. However, in our case, the solitons are stable even at very large voltages (larger than 60 V at $f=100\text{ Hz}$, cell gap $\sim 10\ \mu\text{m}$). In addition, the stability of the solitons in our case is frequency dependent (both homogeneous and homeotropic cases). The threshold voltage of the generation of the solitons is dependent on the frequency, which increases by increasing frequency (Fig. 1c and figure below). And the solitons disappear at fixed voltage by increasing the frequency only, for instance the solitons are stable at $U = 30\text{ V}$, $f = 100\text{ Hz}$ (Fig. 11), but disappear at $U = 30\text{ V}$, $f = 300\text{ Hz}$ (in homeotropic cells). This frequency dependent stability does not make sense if the solitons are torons since torons are topologically protected. Changing frequency does not change their director configurations and destroy their topological structures (ref [43]). Instead, the frequency dependent stability of the solitons is in accordance with “dissipative solitons” reported before (refs [44-50]).

Voltage thresholds of directrons (black squares) and Fredericksz transition (red circles) in cells of homeotropic alignment ($p \sim 10 \mu\text{m}$, $d \sim 9.4 \mu\text{m}$).

Furthermore, the conductivity of the LC material is also very important to the formation of the solitons. The solitons only appear in LCs with relatively low conductivity (ref [44-46, 49, 50]). As the reviewer required below, to measure the existence of fluid flows, we doped a small amount of a fluorescent dye (~ 0.04 wt%) into the sample. However, no solitons are generated in the sample. Instead, a 2D grid convective pattern is observed. We believe this is because the doping changes the conductivity of the LC host. Such a conductivity dependence of the formation of the solitons cannot be explained if the solitons are torons or skyrmions. In fact, we also induced the torons in the homeotropic cell by heating the sample into the isotropic phase and then cooling it back to the nematic phase (Figure below). The figure below shows the sample before (a) and after (b) being applied with an electric field ($U = 40$ V, $f = 100$ Hz). The white dashed line represents the edge of the ITO electrode. On the left side of the line is the region covered with ITO electrode. As the reviewer can see, the directrons ((b), left region) are very different from the torons ((b), right region). Thus, in conclusion, we do not think our solitons are torons or skyrmions. We have also added a short discussion in the “**Circular collective motion of solitons commanded by topological defects**” section to demonstrate that our solitons are not torons or skyrmions.

POM images of torons and directrons. The white dashed line represents the boundary of the ITO electrode. On the left side of the line is the region covered with ITO electrode. Scale bars 100 μm .

As the reviewer suggested, we made samples with different pitches ($p = 2 \mu\text{m}$, $5 \mu\text{m}$, $10 \mu\text{m}$) and different cell gaps ($d = 5 \mu\text{m}$, $10 \mu\text{m}$, and $20 \mu\text{m}$). However, as we mentioned above, because we do not have a confocal microscope and the only equipment in our lab which can characterize the structure of the solitons is a polarizing microscope, so the samples are characterized by polarizing microscope only. The POM images of the solitons in different samples are shown below. As we mentioned above, because the thick glass substrates of the LC cells make focus difficult and the light intensity of our microscope is not strong enough, the images are not very clear. However, we can still see that the textures of the solitons in different samples are similar to each other. The nucleation and dynamics of the solitons are also similar to each other, except that the solitons in sample of large pitch (c) which move in specific directions as reported before (refs [44-46, 49,50]). The solitons in (d) can hardly be observed by our microscope with the higher NA objective (insets). There are two kinds of solitons in cells of $d = 20 \mu\text{m}$ (e and f). At relatively low voltages (e), both the texture and dynamics of the solitons are basically similar to the ones in (a). At higher voltages

(f), solitons with a larger size emerge. These solitons move more quickly and absorb the small solitons shown in (e). These solitons show an emergent “turbulent” collective motion manifested with vortices and jets, which has been briefly demonstrated in the manuscript. All of these solitons show similar properties and we think none of them are torons or skyrmions due to the reasons we demonstrated above.

POM images of solitons in different LC sample. (a) $p = 2 \mu\text{m}$, $d = 9.3 \mu\text{m}$ at $f = 100 \text{ Hz}$, $U = 20 \text{ V}$. (b) $p = 5 \mu\text{m}$, $d = 9.6 \mu\text{m}$ at $f = 100 \text{ Hz}$, $U = 40 \text{ V}$. (c) $p = 10 \mu\text{m}$, $d = 9.4 \mu\text{m}$ at $f = 100 \text{ Hz}$, $U = 40 \text{ V}$. (d) $p = 2 \mu\text{m}$, $d = 5.1 \mu\text{m}$ at $f = 200 \text{ Hz}$, $U = 14 \text{ V}$. (e) $p = 2 \mu\text{m}$, $d = 20.8 \mu\text{m}$ at $f = 200 \text{ Hz}$, $U = 56 \text{ V}$. (f) $p = 2 \mu\text{m}$, $d = 20.8 \mu\text{m}$ at $f = 200 \text{ Hz}$, $U = 64 \text{ V}$. Scale bars are $10 \mu\text{m}$. The insets in (d) show the POM images obtained by a higher NA objective. The white arrows show the polarizers and the yellow arrow show the optic axis of the λ -plate.

Q 1.3 Is there fluid motion or just rotational director dynamics in the experiment? Did authors use tracer particles to detect flows and how these flows correlate or not with motion directions of single or many solitons? I note that a recent theory by Selinger shows how flows are not necessary for soliton motions: <https://arxiv.org/abs/2109.07314> - confirming this would be both timely and important, but of course this could be different for what authors study. If authors have some flows - is it just flow carrying solitons, with material transport involved in the forms of flows, and the solitons along with that? If so, is it still interesting? Certainly, this might not be as emergent and relevant to active matter in this case as all authors would probe would be these fluid flow currents and how smaller micro-rivers combine into bigger ones, not the collective self-propulsions. Authors should explore how the flows and mass transport, if existent, correlate with soliton and collective soliton motions.

A 1.3 We thank the reviewer for his/her comments. To detect the potential fluid flows, we firstly doped a very small amount of a fluorescent dye (0.04 wt%) into the LC system. However, as we mentioned before, the formation of the solitons is closely related to the conductivity of the LC material and the fluorescent dye usually greatly changes the conductivity of LCs. As a result, we

did not see any solitons, instead a global 2D grid convective pattern (figure shown below) is observed in the doped system.

We then doped micro-particles (diameter 3 μm) into the LC system. The doping slightly decreases the voltage threshold of the solitons. We find that individual micro-particles are very easily pinned on the glass substrates of the cell by applying the electric field. On the other hand, some aggregates of micro-particles and dusts do move at relatively high voltages. However, it is always observed that there are solitons attached on those. The solitons always firstly nucleated at particles or dusts and then move them. Such a behavior has also been reported by Li et al. (ref [92]) and is called “soliton-induced liquid crystal enabled electrophoresis”. So it is not clear whether the particles are moved by the solitons or by some fluid flows, or both. However, by slightly decreasing the voltage, the solitons near the aggregates and dust inclusions disappear and the aggregates and dust stop moving. As the reviewer can see from the figure below, by increasing the applied voltage from $U = 10$ V to $U = 12$ V, the director deformation around the dust particle increases (demonstrated by the increase of the transmitted light intensity). The particle then starts moving. By decreasing the voltage back to $U = 10$ V, the director deformation relaxes and the particle stop moving. We think that this behavior can be attributed to the accumulation of space charges at the surface of particles and dust inclusions, which induces the director deformation around the particles and leading to the formation of the directorons. Such a behavior is in accordance to the cargo-transport reported by us previously (ref [46]).

We then doped quantum dots (0.005 wt%) into the LC media. The voltage thresholds of the solitons are greatly decreased by the doping and once the voltage is larger than 20 V, electro-convection patterns emerge and cover the whole sample. We observed the sample with fluorescent microscopy (borrowed from other labs), however, nothing was observed. We then increased the concentration of quantum dots to 0.1 wt%. At such a high concentration, no soliton was generated and the sample was again covered with electro-convection patterns. Although the concentration is too high to induce the solitons, we could still observe nothing under the fluorescent microscope, except a very weak blueish background. This maybe because the quantum dots are too small to be seen.

In conclusion, we tried different methods to consider the existence and the influence of fluid flows, however, due to different reasons mentioned above, no conclusive conclusion about the role of fluid flow can be made. We believe there is no obvious fluid flows before the emergence of the solitons since the micro-particles and dust inclusions do not move until the emergence of solitons. After the emergence of the solitons, there could be some fluid flows which are either generated by the motion of solitons or other factors such as ion motion. But we do not think that the solitons are simply carried by some kind of fluid flow. Since the dielectric anisotropy of the LC material is negative, there is no backflow generated by the rotation of the director field. No electro-convection pattern is observed which can exclude the electro-convective flows. The isotropic flows generated by the injection of ions usually occur at very low frequencies. However, our solitons can move effectively at relatively high frequencies (up to 800 Hz or even higher). The only possibility left is the flow generated by ion motion. However, this flow also usually happens at relatively low frequencies, i.e.,

the conductive regime, which is limited from above by the critical frequency $f_c = \sqrt{\zeta^2 - 1} / \tau_M$

(E. Dubois-Violette, et al., J. Phys. Fr., 32, 305-317, 1971). $\tau_M = \epsilon_0 \epsilon_{\perp} / \sigma_{\perp}$ is the Maxwell

relaxation time for planar cells, $\varepsilon_0 = 8.85 * 10^{-12} \text{ Fm}^{-1}$. $\zeta^2 = \left(1 - \frac{\sigma_{\perp} \varepsilon_p}{\sigma_p \varepsilon_{\perp}}\right) \left(1 + \frac{\alpha_2 \varepsilon_p}{\eta_c \Delta\varepsilon}\right)$ is the

material parameter that depends on conductivities (σ_{\perp} and σ_{\parallel}), permittivities (ε_{\perp} and ε_{\parallel}), and viscous coefficients (α_2 and η_c). Using the material data (Methods), $1/\tau_M \sim 89 \text{ Hz}$. The factor

$\sqrt{\zeta^2 - 1}$ is hard to determine exactly since both α_2 and η_c are not known. $\left(1 - \frac{\sigma_{\perp} \varepsilon_p}{\sigma_p \varepsilon_{\perp}}\right) \sim 0.875$

and $\frac{\varepsilon_p}{\Delta\varepsilon} \sim -0.65$. We assume the ratio $-\frac{\alpha_2}{\eta_c}$ being on the order of 1 as Li et al. did in their work

(ref [45]), the critical frequency $f_c \sim 128 \text{ Hz}$. However, as we mentioned before, the solitons in our case can move effectively at frequencies up to 800 Hz. So, as far as we know, it seems that there is no such kind of flows which can carry the solitons in random directions and lead to the collective behavior. However, to conclusively find out about the existence and the influence of the flows, further systematic investigations are required which currently cannot be done in our lab due to limited equipment and techniques. We have added a short discussion in the second part of the results section to discuss the possibility of the existence of the flows.

Grid convective pattern in fluorescent dye doped LC system at $U = 16 \text{ V}$, $f = 100 \text{ Hz}$.

A directron nucleated at a dust particle and moves it. Scale bar 50 μm .

Q 1.4 Page 1, 1st paragraph – saying that “so far great achievement... has been achieved through numerical modeling...” – this is not accurate now (statements like this were appropriate 10 years ago) as active matter is the most active branch of the soft side of condensed matter now and hundreds of outstanding experiments and analytical models are reported every month. Authors should refer to many elegant experimental and theory works by Dogic, Fraden, Irvine, Vitelli, Yeomans, Bowick...

A 1.4 We thank the reviewer to point out the inappropriateness of our expression in the introduction. We have changed it to “However, it is difficult to study collective behavior by directly performing quantitative measurements in conventional macroscopic systems such as mammal herds, where tracking individual motions of a large population over long periods of time is extremely challenging. As an alternative, great achievement has been made through numerical modeling⁸⁻¹². Moreover, different kinds of micro-scale experimental systems have also been developed to utilized as active models for studying collective behavior¹³⁻¹⁵.” We have also added a short discussion in the discussion section to refer the experimental and theory works the reviewer mentioned.

Q 1.5 Authors refer to their solitonic structures as bullets, but the original historic name for these solitons was “butterfly”, given by Cladis. I have seen some other names, like “directron”. Perhaps there could be a good reason to give a different name once the structure is known, but so far the “baterfly” name seemed to be consistent with some images in nematics. The work by Cladis and her team [Phys. Lett. A 235, 508 – 514 (1997).] should be referenced as the first experimental study of such solitons. I note that authors were kind of critical and looking down on this work (and this style I would discourage) in their Comp Phys article, saying “It should be noted that a similar phenomenon was earlier observed by Brand et al.³², who reported localized formations in the shape of “butterflies” that could move in the plane of the cell. However, neither the director structure ... was revealed in their report.” I must say that, unfortunately, this structure in both planar and homeotropic cells that authors study is still unknown as well, though I think the solitons in homeotropic cells are torons. 30 years after Cladis’ work, authors have more tools to uncover the structures of solitons they study. Authors say “Different kinds of solitons have been

produced in LCs. However, most of them are immobile” - I would disagree as I have seen books & hundreds of articles on mobile liquid crystal solitons, many more than what was devoted to static solitons.

A 1.5 We thank the reviewer for his/her comments. Indeed, as the reviewer said, different kinds of names, such as, director bullet, directron, butterfly, etc. were used to name the solitons. The “director bullet” is actually coined by Li et al. (ref [44]), they also call the solitons as “directron” (ref [50, 92]). In the manuscript, we did not really name our solitons as “bullets”. In our previous work, we mentioned that the solitons in chiral nematics looks like “bullets” and the ones in achiral nematics look like “butterflies” (ref [46]). To avoid further confusion, we have now named the solitons as “directrons” in the manuscript according to refs [50, 92].

As the reviewer suggested, we have referenced the work by Cladis and her team [Phys. Lett. A 235, 508 – 514 (1997)] as the first experimental study of the solitons. We are definitely not critical and looking down on this work, on the contrary. We would like to apologize if we made this impression on the reviewer or any other reader. We also agree with the reviewer that the structure of the solitons is also not clearly unveiled in our work. However, unlike the reviewer suggested, in fact, we do not have more tools to uncover the structure of solitons compared to what Cladis et al had available. The fact that the structure of the solitons has been unclear for about 30 years also demonstrates its difficulty and complexity. To totally unveil the structure of the solitons, especially the chiral ones, further experimental and theoretical investigations are required.

The mobile LC solitons, as far as we know, includes mobile walls in nematics (L. Leger, Sol. St. Comm., 10, 697-700, 1972), director waves in shearing nematics (G. Z. Zhu, Phy. Rev. Lett., 49, 1332, 1982; L. Lin, et al., Phy. Rev. Lett., 49, 1335, 1982; etc.), ring-shaped solitons in pressed nematics (C. Q. Shu, et al., Liq. Cryst., 2, 717-722, 1987; S. Zheng, et al., Phy. Rev. A, 38, 5941, 1988), localized convective rolls in nematics (A. Joets and R. Ribotta, Phy. Rev. Lett., 60, 2164, 1988; D. Igner and J. H. Freed, J. Chem. Phys., 76(12), 6095, 1982; J. H., Huh, Phy. Rev. E, 95, 042704, 2017; etc.), The topological soliton in chiral nematics, such as skyrmions and torons (Refs [25, 42, 43], etc.), and the directrons in this manuscript. We are sorry for our inappropriate expression and have changed the sentence into “Different kinds of solitons have been produced in LCs³³, both immobile³⁴⁻³⁸ and mobile ones^{24, 39-42}.”

Q 1.6 Authors study all kinds of collective behaviors and refer to herds and swarms and similar formations of particles that only interact through bumping into each other, like in some of the original active matter toy models. However their medium is a liquid crystal, the soliton structure in director field implies elastic energy costs - here must be elastic interactions between them - what is their role? Recent work from LosAlamos even predicts spin ice formation in solitons due to such interactions: 10.1103/PhysRevLett.126.047801 , DOI: 10.1039/b000000x If there are flows, are there hydrodynamic interactions as well? What is the Ericksen number? I urge authors to probe what is the nature of interactions between solitons? Elastic, hydrodynamic, electrostatic?

A 1.6 We thank the referee for the comments. Yes, since the solitons are actually localized director deformations, there are elastic interactions between them which show repulsive force at short range but attractive force at relatively long range (Fig. 2e). The long-range attractive force leads to the formation of soliton flocks and the short-range repulsive force makes the solitons behave like elastic

particles and aligns them during collisions, leading to various collective behaviors. When the solitons move through the nematic bulk, they may induce some hydrodynamic flows and there may be hydrodynamic interactions between the solitons, which can be one of the factors for the emergence of various collective behaviors (especially for the “turbulent swimming pattern” shown in Fig. 11). Because we cannot measure the fluid flow, it is difficult to calculate the Ericksen number exactly. However, as we discussed before, before the formation of the solitons, we think there is almost no fluid flow, so the Ericksen number is close to 0. The velocity of the solitons is dependent on the applied electric field. We think that in cases of low soliton density and small soliton velocity, since the solitons gradually form flocks through elastic interaction and no turbulent flow pattern is observed, the Ericksen number is also negligible.

Q 1.7 Authors refer to the texture in homeotropic cell as corresponding to umbilics when in-plane component of field is induced in hom. cells. However, it could be also that authors create the so-called dowser texture (see many elegant works by Pieransky & others): <https://onlinelibrary.wiley.com/doi/abs/10.1002/9781119850809.ch4> At least the possibility should be discussed and authors should say how they checked. The umbilic in this case could be a hedgehog point defect, which might be explaining the fact that it becomes a generation site for the solitons seen in one of the movies...

A 1.7 We thank the reviewer for the comments. As the reviewer pointed out, the POM images of dowser texture and umbilic defects are similar. However, the structure of umbilic defects is not singular, the director within the umbilic defects continuously tilts toward the substrate normal, being parallel to it at the core of the defects (A. Saupe, *Mol. Cryst. Liq. Cryst.* **21**, 211, 1973; I. Dierking, et al., *Phys. Rev. E*, **71**, 061709, 2005). As a result, the size of the core of umbilic defects can be continuously changed by tuning the applied electric field. On the other hand, the core of the dowser texture is a hedgehog point defect, which can only be transformed to a looped line defect (J. M. Gilli, et al., *Liq. Cryst.*, **23**, 619-628, 1997). In our experiments, the size of the core of the defects can be continuously changed by tuning the applied voltage (Figures below), so we think these are umbilic defects. As the reviewer suggested, we have added a short discussion to demonstrate that the defects are umbilic defects not the dowser texture in the “**Circular collective motion of directrons commanded by topological defects**” section.

POM image of a pair of umbilic defects at varied voltages. Scale bar 100 μm .

The voltage dependence of the diameter (D) of the umbilic defect core. The red circles represent the $s = +1$ defects and the black squares represent the $s = -1$ defects. The inset shows the POM images of an umbilic defect. The diameter is measured as the width of the cross-section of the defect core as indicated by the yellow lines. Scale bar 10 μm

Q 1.8 Page 2, top paragraph – authors motivate their study by noting that the prior work on soliton dynamics had a faster synchronization time scale, but they prefer longer than seconds. Indeed, movies in the present article seem to be 10X sped up, so the system is slow indeed. However, it would seem that this new study would be an incremental accomplishment if this was the only progress made as compared to that old NatCom paper. Moreover, the same author, H. Sohn, and other authors have done much more in the field of active/driven solitons (PNAS 117, 6437 (2020), Opt Express 28, 6306 (2020), Opt. Exp. 27, 29055 (2019), PR E 97, 052701 (2018)···), so authors should do a more thorough review of this literature and better describe the novelty of what they do to merit publication in NatCom. Moreover, some of these other papers had regimes even closer to what authors study here, including slower synchronization. The present study indeed seems to be VERY inspired by the works of Sohn as it reports similar types of characterizations that Sohn did in his series of papers, as well as characterization of the same effects (velocity order parameters, giant number fluctuations as in Nat Comm, motions around induced umbilical defects in the OptExp. Article, hexatic like ordering in PNAS···). On top of this, as I already mentioned, I believe some of the solitons authors study are the same torons that Sohn studied in cells of similar thickness, pitch and same material ZLI-2806... It does not mean authors have no novel results – they clearly see something different too, which should be highlighted and put on shelves distinguishing prior art from what is done in this work.

A 1.8 We thank the reviewer for his/her comments. We are familiar with all the publications the reviewer mentioned. However, most of the works mentioned by the reviewer are mainly about electrically driven motion of skyrmions. For instance, the work (PRE 97, 052701 (2018)) is mainly about motion of individual skyrmions and cargo transport. Since our solitons are not torons or skyrmions, it's not really related to our current work. The work (Opt. Exp. 27, 29055 (2019)) is mainly about tuning the structure of skyrmions and their assemblies through optical manipulations, which we think is also not very related to collective behavior. The work (PNAS 117, 6437 (2020)) is mainly about the 2D crystal structure formed by a lot of torons and studying the evolution of the crystal structure and the motion of the defects within the crystal through the motion of the torons. Although there are many moving torons, the paper does not talk about the collective motion, so we do not think it is directly related to the present manuscript. The work (Nat. Com. 10, 4744 (2019)) is surely very relevant to the manuscript and indeed, as the reviewer said, our work is, to some extent, inspired by this work. However, there are great differences between this work and ours. First and most importantly, as we mentioned before, the solitons in their system and our system are different kinds of solitons. Secondly, the skyrmions in their systems only show limited types of collective behavior. All the skyrmions tend to synchronize their motions immediately and move together at a constant velocity and in the same direction. The work (Opt Express 28, 6306 (2020)) is also relevant to our work, in which more complicated collective behaviors are realized through optical manipulations. However, again, the solitons are skyrmions which are different from ours, and the setup of the experimental system is relatively complicated which includes optical manipulations tools, such as laser tweezers, and photosensitive chiral dopants. Moreover, collective behaviors that can be observed in our system and many other active systems such as self-organization of flocks, fission and fusion process, density dependent collective motion, swirling and vortices (which was observed after we submitted the manuscript and has been added into the manuscript now), etc. are not realized.

We did use the same characterization techniques, such as velocity order parameter, giant number fluctuations, as Sohn et al. did in their work (Nat. Com. 10, 4744 (2019)). However, these techniques are also broadly used in many other investigations of active systems simply because they are very adequate (A. Bricard, et al., Nature, 503, 95, 2013; Nat. Com., 6, 7470, 2015; V. Narayan, et al., Science, 317, 105, 2007; J. Deseigne, et al., Phys. Rev. Lett., 105, 098001, 2010; etc.). We also characterized our system with spatial and temporal correlation functions which are not used in Sohn's works but are broadly used in many other works of active systems (J. Toner and Y. H. Tu, Phys. Rev E, 58, 4828, 1998; H. P. Zhang, et al., PNAS, 107, 13626-13630, 2013; V. Schaller and A. R. Bausch, PNAS, 110, 4488-4493, 2013; X. Chen, et al., Phys. Rev. Lett., 108, 148101, 2012). Yes, the guiding of the motion of the solitons by umbilic defects is also reported in (Opt Express 28, 6306 (2020)). However, unlike the directrons here which are trapped and persistently swirling around the $s = +1$ defect but repelled by the $s = -1$ defect, the skyrmions, in that work are only deflected and sidetracked by the umbilic defects. In addition, the umbilic defect in that work is unpaired and induced by the assistance of optical manipulation.

The hexatic ordering of torons in the work (PNAS 117, 6437 (2020)) is also different from the "hexatic phase" that we mentioned in the manuscript. The hexatic phase is a thermodynamic phases of equilibrium 2D systems predicted by the Kosterlitz-Thouless-Halperin-Nelson-Young (KTHNY) theory, which is located between the isotropic liquid phase and the crystal phase. The authors did not talk about this in the work (PNAS 117, 6437 (2020)). What they were mentioning in that

publication was the hexagonal arrangement of the torons. We have deleted the discussion about the hexatic phase in the rectified manuscript because it is not relevant to the current topic and in order to better compare our system with other active systems, we now compute spatial correlation functions in a “local coordinate frame” (Fig. 3a inset), which eliminates the sixfold angular symmetric diffraction pattern of the correlation functions (which usually indicates the formation of the “hexatic phase”) (Fig. 6). More details about the “hexatic phase” will be discussed in another publication.

To conclude, we do not think that our work is an incremental accomplishment of the works published by Sohn et al. Instead, we believe our system behaves more like real living systems and exhibits more intriguing collective behaviors. We have added a short text in the discussion section to demonstrate the difference between their works and ours.

As the reviewer suggested, we have rectified the manuscript to highlight our findings and distinguish our work from other works.

Q 1.9 In summary, I cannot recommend the manuscript in its present form, but I will be happy to look at responses and revisions.

A 1.9 We again thank the reviewer for evaluating our work thoroughly and giving us many helpful suggestions and comments to help us improve our manuscript. We hope that we have answered all the reviewer’s questions and solved the problems that he/she discussed. We believe that the rectified manuscript is now adequate for publication.

Reviewer #2 (Remarks to the Author):

Q 2.1 The manuscript demonstrates rich collective behavior of 3D director solitons in a chiral nematic but the presentation makes it difficult to recommend publication.

A 2.1 We thank the reviewer for spending his/her precious time on evaluating our work. We have rectified the manuscript based on the reviewers’ comments and suggestions. We believe that the manuscript is now adequate for publication.

Q 2.2 - What are the noteworthy results?

A. Observation of random trajectories of solitons that arrange in coherent structures and measurement of giant density fluctuations are noteworthy results; however, lack of experimental characterization makes it difficult to understand the underlying mechanisms

A 2.2 We thank the reviewer for his/her comments. We have added more experimental characterizations about the soliton system and added more discussions about the formation mechanism of the solitons in the manuscript. We have also described a new collective behavior of the solitons which was observed after submitting the manuscript. We have rectified the results section to make it complete and easier to understand. We have further improved the manuscript to highlight our findings and the importance of our presented work.

Q 2.3 - Will the work be of significance to the field and related fields? How does it compare to the established literature? If the work is not original, please provide relevant references.

B. The observations are new and interesting, but the presentation lacks depth on substance and mechanisms; comparison with other active systems are not always justified

A 2.3 We thank the reviewer for his/her comments. We have added more experimental characterizations and discussions in the manuscript and rectified our description and expression.

Q 2.4 - Does the work support the conclusions and claims, or is additional evidence needed?

C. Additional evidence is needed. The experimental results are described in a manner that makes it hard to understand the conclusions. For example, the manuscript does not even specify the concentration of the chiral dopant in the mixture. Furthermore, the anisotropy of dielectric permittivity and electric conductivity are important factors in electrohydrodynamics of liquid crystals in general and the formation of solitons in particular. However, the manuscript does not discuss the role of these anisotropies; the data on anisotropies are relegated to the end, making the main part of the text hard to understand. There is no discussion of how the length and width of the solitons affect the correlation functions and how the maxima and minima of these are related to the length and width of the solitons. It is not clear how the authors established that the director oscillates with the frequency of the applied field. It is not clear how the director distortions shown in Fig 2a were established.

A 2.4 We thank the reviewer for his/her comments. We have added a description in the Materials section to demonstrate the concentration of the chiral dopant as “The weight concentration of chiral dopant is ~ 6 wt% in chiral nematic mixture of $p \sim 2 \mu\text{m}$ and ~ 1.2 wt% in chiral nematic mixture of $p \sim 10 \mu\text{m}$.”

We have also added a discussion in the result section “**Formation of directrons**” to demonstrate the formation mechanism of the solitons and the important roles of the dielectric permittivity and conductivity played in the formation of the solitons.

The solitons in our experiment are basically circular in the xy plane and therefore there is no width or length. We are not quite sure what does the reviewer mean by saying that “There is no discussion of how the length and width of the solitons affect the correlation functions and how the maxima and minima of these are related to the length and width of the solitons?” Does he/she mean that we should normalize the spatial correlation functions with the size of solitons? If so, we can surely do that.

We demonstrate that the director within the solitons oscillates with the frequency of the applied electric field according to our previous work on the same kind of solitons (ref [46]) and the work by Li et al. (refs [44, 45]). In those works, the dynamic structure of the solitons in achiral nematic is characterized. The intensity of the light transmitted through the solitons changes linearly with the frequency of the applied electric field. In this work, the solitons are in chiral systems, making the characterization methods used in previous studies inappropriate. Moreover, the solitons here do not occur at frequencies lower than 20 Hz. The limited frame rate of our camera and the limited light

intensity of our microscope (Higher frame rate requires shorter exposure time, and thus larger light intensity. If the frame is higher than 200 fps, the image will be very dark and one cannot see anything.) make it difficult to characterize the director oscillation as we did in ref [46]. However, we still managed to get the time sequence of POM images of the solitons at $U = 20$ V, $f = 40$ Hz. The figure is added as Supplementary Fig 3. In the figure, although the images are not very clear, one can still see that the soliton structure changes periodically with the frequency of the applied electric field. We have added more discussions about the oscillation of director in the first and second parts of the result section.

Due to the small pitch of our LC system and the limited resolution of light microscopy, the distortion of the soliton structure demonstrated in Fig. 2a cannot be clearly observed. However, according to our previous studies and works by other groups (refs [44-50]), the motion of the solitons is determined by the symmetry breaking structure of the solitons as indicated in Fig. 2a. The distortion of the soliton structure in Fig. 2a is thus deduced based on the previous studies (refs [44-50]) and the structure of the solitons obtained from the POM images (Supplementary Fig. 2). We have added more details to demonstrate this distortion and its relationship with the motion of the solitons in the second part, “**Dynamics of directrons**”, of the results section.

Q 2.5 - Are there any flaws in the data analysis, interpretation, and conclusions? - Do these prohibit the publication or require revision?

D. The presentation misses important points relevant to the system at hand (director field, its oscillations with the frequency of the applied field, the role of anisotropies, size of solitons, plausible mechanisms of interactions, etc.) and instead draws superficial links to other systems, such as flocks of birds.

A 2.5 We thank the reviewer for his/her comments. We have added more characterizations and descriptions about the experimental system in the manuscript. The missing points (director field and its oscillation, the role of dielectric and conductivity anisotropies, size of solitons, interaction mechanisms, etc.) have been added into the manuscript. The comparison with other active systems are demonstrated with more details.

Q 2.6 - Is the methodology sound? Does the work meet the expected standards in your field?

E. It is not clear how the soliton trajectories are related to the local director on the cholesteric pseudo-layers. It is not clear how the director field configuration was established. It is not clear how the oscillation frequency of the director was measured.

A 2.6 We thank the reviewer for his/her comments. We are not sure what the reviewer means by “It is not clear how the soliton trajectories are related to the local director on the cholesteric pseudo-layers.” We presume that he/she means how Fig. 2a is established. This we have explained in **A 2.4**. The director field configuration, as we mentioned above, is deduced from the POM images and the formation mechanisms of the solitons. More details about it have been added into the manuscript (Supplementary Fig. 2). The question about the oscillation of the director field within solitons has also been explained in **A 2.4**.

Q 2.7 - Is there enough detail provided in the methods for the work to be reproduced?

F. No, please see above.

A 2.7 We again thank the reviewer for evaluating our work and his/her comments. We have added considerably more details about our experimental system in the methods section and believe that sufficient information is now provided for other groups to reproduce the experimental work. We hope we have now answered and resolved the problems pointed out by the reviewer.

Q 2.8 Other comments:

1. The Abstract must describe the system under study. The current text is too general to understand what the solitons are and which material they form in and under which circumstances. Terms such as “animal fluid” are ambiguous.
2. The text relies heavily on the very general term “soliton” . Although it is true that the observed objects are solitons, this general term is not descriptive enough of the findings, since there are many different solitons in Nature and laboratories. Why not use more specific terms?
3. Statements such as “the director is uniformly self-assembled into a helical superstructure” are not scientifically sound. Self-assembled might be, for example, molecules or colloids, but the director is in a different category.
4. What is the nature of interactions of solitons along the normal to the plates and in the plane of the cell?
5. What is the origin of a “cutoff size” in the system?
6. Why in some cases the increase of the field leads to more ordered structures and in other cases to chaotic behavior?

A 2.8 We thank the reviewer for his/her comments and questions. Below are our answers to the questions:

1. We have rectified the abstract as the reviewer suggested. The term that we used in the abstract is “animate fluid” not “animal fluid”.
2. As the reviewer suggested, we have now used the term “directron” to name the solitons observed, as suggested by the authors in ref [50, 92].
3. We have changed the corresponding sentence as “Along the normal to the glass substrates, the director twists continuously along a helical axis at a constant rate.”
4. The system is basically 2 dimensional, so we do not think there is any interactions along the normal to the plates. Since the solitons are director deformations, we believe the interaction between solitons is basically elastic interaction. The solitons show a short-range repulsive and a relatively long-range attractive interaction, Fig. 2e. This is further detailed and discussed in the revised manuscript.
5. The group size distributions usually fit a truncated power law with a crossover to an exponential decay (refs [69, 70]). The cutoff size characterizes the transition from power law to exponential behavior of the group size distribution (I. D. Couzin and J. Krause, Self-Organization and Collective Behavior in Vertebrates, Advances in the Study of Behavior, Vol. 32). In our experiment, the cutoff size originates from the fission-fusion process of the soliton groups, i.e.,

the soliton groups are not stable. They change their size (number of solitons within a single group) during motions. It also depends on the interactions between solitons, the soliton density, and the dynamics of the solitons.

6. The increase of the field leading to more order structures happens at relatively low voltages. This is basically due to the increase of the soliton density. At low soliton densities, the soliton flocks move in different directions. By increasing the density, the small flocks collide and combine into large flocks, within which the solitons move in the same directions. Finally, at the voltage of 18 V, the whole sample is filled with solitons which move coherently in the same direction. Such a phenomenon has been observed in many active systems (A. Sokolov, et al., *Phy. Rev. Lett.*, 98, 158102, 2007; J. Deseigne, et al., *Phy. Rev. Lett.*, 105, 098001, 2010; etc.). We have added this explanation into the manuscript. The chaotic motion of the solitons usually happens at relative high voltages. Such a behavior is due to the increase of the background noise of the system, which can be caused by different reasons, such as the increase of the velocity of the solitons, the distortion of the soliton structures, the ion injection, the electrohydrodynamic effect, etc. Such a phenomenon is also observed in other active systems, such as granular systems in which the systems show an order-disorder transition by increasing the amplitude of the vibration of the system (J. S. Olafsen and J. S. Urbach, *Phy. Rev. Lett.*, 95, 098002, 2005). We have also added the explanation into the manuscript. The behavior is similar to the description of the Vicsek model, in which ordered dynamic patterns arise by increasing the density but disordered motions arise from increasing the noise of the system.

Reviewer #3 (Remarks to the Author):

Q 3.1 The manuscript entitled "Electrically tunable collective motion of dissipative solitons in chiral nematic films" by Yuan Shen and Ingo Dierking presents a collective motion of chiral nematic solitons. This is a kind of follow-up of the works on the nematic solitons reported by Li et al. and by the present authors. In this paper, the authors found that the motions of nematic solitons mimicking the collective behaviors established in many active matter systems, such as flocks of birds or schools of fishes. This concept works very well and may attract broad readership in the field of physics. especially at this timing that Prof. Parisi won this year's Nobel prize for physics. The experimental results in this work are well examined. Data processing is properly made and the consequence is scientifically sound. So basically I think this is a nice paper and should be published in Nature Communications. My only concern is, most of explanation in this paper is just qualitative consideration (only size distribution in Fig.5 is discussed well compared to others), and thus there is slight lack of quantitative physical analysis based on formulation. For instance, one of the intriguing points in the active matter physics is scale-free correlations like the flocks of birds. The authors have summarized the 2D correlation functions and their behavior on the relative coordinate r . So, the rescalability among the soliton flocks might be interesting to be discussed further in these data upon comparison with other scale-free systems.

A 3.1 Firstly, we would like to thank the reviewer for spending his/her precious time on evaluating our work. We also thank the reviewer for the very positive evaluation of our work. We have added more experimental characterizations and theoretical explanations about the system in the manuscript.

To better compare our soliton system with other active systems, such as bacteria suspensions, we have computed the spatial correlation functions in a “local coordinate frame” (Fig. 3a inset) as other authors did in different active systems, such as bacteria suspensions, driven filament systems, active colloidal systems, etc. (H. P. Zhang, et al., PNAS, 107, 13626-13630, 2010; V. Schaller and A. R. Bausch, PNAS, 110, 4488-4493, 2013; J. Zhang, et al., Nat. Phys., 17, 961-967, 2021). As a result, the spatial velocity correlation function shows an anisotropic profile (Fig. 3b) which is also observed in many other active matter systems. We have also added more characterizations of our soliton system, such as temporal velocity correlation function, turbulent swimming pattern of solitons, etc. into the manuscript and compared the correlations of our system with those of bacteria suspensions.

Q 3.2 More technically, I have several comments as follows;

1. If I understand correctly the experimental condition, the director field of the present system is twisting along the cell normal. Then, it is difficult to imagine how the solitons as schematized in Fig.1d can be distributed with twisting in the cell? It would be appreciated if the authors could improve this point by drawing the exact director distortion in a soliton in a better way (cross-sectional views may work, in my opinion).

A 3.2 We thank the reviewer for his/her comments and suggestion. We have discussed the formation mechanism of the solitons and deduced the soliton structure from the POM image which has been added into the manuscript as Supplementary Fig. 2. Unfortunately, we could not give more experimental characterizations and evidence (such as confocal microscopy, etc.) of the structure of the solitons due to the limited availability of such equipment. No such instrument is available in our department.

Q 3.3 2. It would be also nice, if the initial director observation is described somewhere in the manuscript or in the supplementary information. The used cell was a planer cell with a homogeneous surface condition but no given preferable alignment direction. In such a case, I guess there are lots of line defects which may trap solitons (flocks).

A 3.3 We thank the reviewer for his/her suggestion. As the reviewer suggested, we have added an POM image of the initial texture of the sample without applying electric field as an inset in Fig. 1c. Actually the inside surfaces of the homogeneous cells (maybe we should call it planar cells) used in the experiment are spin coated with polyimide and rubbed in a specific direction. This was an unclear description about the anchoring condition on our part. We have now added more details about the alignment condition and the initial director structure in the first part, “Formation of directrons”, of the result section.

Reviewer #4 (Remarks to the Author):

Q 4.1 The authors present a number of interesting experiments, where large numbers of solitons in a chiral nematic liquid crystal are created by applying an external electric field of varying amplitude and frequency. The authors find that these solitons resemble flocks in active matter systems, because they experience collective motion, they are able to synchronize their motion and some eye-catching

dynamic phenomena of schools of solitons are seen on the videos.

This work is without doubt very interesting and novel. This work is also timely, as active nematics are a topic of great current interest. In this sense this work is quite unique, because there are not many experimental settings that would display the behavior of active matter in liquid crystals. There is no doubt that this work opens a new direction in liquid crystal research, bridging the gap between the liquids and liquid crystal communities. The significance of this work is high.

While the topic presented is of great current interest, the manuscript itself has many deficiencies, which need to be removed before final decision on publication is reached. In particular I have the following questions and comments:

A 4.1 We thank the reviewer for spending his/her precious time on evaluating our work. We also thank the reviewer for his/her positive evaluation of our work by pointing out that “There is no doubt that this work opens a new direction in liquid crystal research, bridging the gap between the liquids and liquid crystal communities. The significance of this work is high.”

Q 4.2 Why is the intriguing collective behavior of solitons observed in this experiment and not in many previous experiments on similar solitons by the same authors? What is the key difference here, in this experiment, compared to many other similar experiments? The authors have published several articles on solitons in chiral nematic liquid crystals, using practically the same LC materials and experimental cells. For example, the authors have studied ZLI-2806 and a chiral dopant ZLI-811 in previous experiments published in *Comm.Phys.* 2020, but have not reported schools of solitons. Is the reason in the chirality, i.e. the length of the pitch? This needs to be clarified to get broader insight into the emergence of active matter behavior of solitons.

A 4.2 As the reviewer suspected, the pitch of the LC media is very important for the emergence of the collective behavior of the solitons. During the experiment, we tried chiral nematics with different pitches, $p = 20 \mu\text{m}$, $10 \mu\text{m}$, $5 \mu\text{m}$, and $2 \mu\text{m}$, respectively. However, we only observed the collective behavior in the chiral nematics of $p = 2 \mu\text{m}$ (homogeneous alignment cells). In the systems of other pitches, the density of the solitons is relatively low and the solitons move either parallel or perpendicular to the alignment direction. Moreover, the solitons can pass through each other like waves, just as reported in previous literature (refs [44-46, 49, 50]). But in the system of $p = 2 \mu\text{m}$, the solitons can move in random directions in the xy plane and behave like true particles that repel each other when they collide. We have briefly explained this in the discussion section.

Q 4.3 On page #2 the authors give a brief and not convincing explanation of why the solitons move "Within the soliton, the director field oscillates with the frequency of the applied electric field due to the flexoelectric effect, breaking the mirror symmetry of the solitons and driving them to move through the uniform nematic bulk." This needs to be better explained, also the structure of a single soliton has to be clearly described, together with the role of dielectric anisotropy and conductivity anisotropy. These are mentioned in section Methods, but their role in soliton propulsion is not clarified.

A 4.3 We thank the reviewer for his/her comments. We are sorry for the brief and incomplete explanation about the formation and motion of the solitons. The brief explanation is basically based on previous works (refs [44-46, 49, 50]). The movement of the solitons is attributed to the periodic oscillation of the director within the solitons. If the symmetry of the soliton structure is broken, this oscillation of the director can then propel the solitons to move. We have added more details to explain the formation and motion of the solitons in the first and second part of the result section. We also discussed the possibility of fluid flows in the system as required by the reviewer #1. We have also done more characterizations on the structure of the solitons and the oscillation of the director field which have been added as Supplementary Fig. 2 and 3. The role of dielectric and conductivity anisotropies in the generation of the solitons has also been discussed in the result section.

Q 4.4 The LC cell structure should be better explained and more accurate. The statement "The experimental setup is similar to that in LC displays, where a thin film of cholesteric LC is sandwiched between two pieces of glass substrates which is coated with an indium tin oxide (ITO) layer as electrodes and a rubbed polyimide layer as the planar alignment layer." is not clear, as the reader might not be familiar with what "planar alignment" is and how the helix is oriented with respect to the cell surface.

A 4.4 We thank the reviewer for his/her comments. We have demonstrated the "planar alignment" and the direction of the helix. The sentence has been changed as "The experimental setup is similar to that in LC displays, where a thin film of cholesteric LC is sandwiched between two pieces of glass substrates which is coated with an indium tin oxide (ITO) layer as electrodes and a rubbed polyimide layer as the planar alignment layer, i.e. the director near the glass substrates align along the rubbing direction. Along the normal to the glass substrates, the director twists continuously along a helical axis at a constant rate."

Q 4.5 What is meaning of "large chirality" in paragraph starting with Dynamics of solitons? Is this the absolute value of the pitch? The authors indicate that omnidirectional movement of the solitons is due to "large chirality" but do not try to explain the reasons for such behavior.

A 4.5 We thank the reviewer for his/her comments. We are sorry for our inappropriate expression. What we mean is "small pitch". The solitons in achiral nematics and chiral nematic with larger pitches move either parallel or perpendicular to the alignment direction. The omnidirectional movement of the solitons observed here is due to the rotational symmetry of the small pitch helical structure of our system. We think that the relatively large ratio of cell gap to pitch of our system suppresses the influence of the rubbing alignment. On the other hand, the director deformation of the solitons reaches maximum in the middle layer of the LC media, and gradually diminishes as one moves toward the top and bottom cell substrates (ref [44]), which further reduces the influence of the rubbing alignment. To prove our speculation, we measured the Brownian motion of a colloidal micro-particle in our system. According to previous studies, micro-particles always exhibit anisotropic diffusion in homogeneously aligned nematics (J. C. Loudet, et al., Science, 306, 1525, 2004; T. Turiv, et al., Science, 342, 1361, 2013; etc.). However, the particles in our systems exhibit an isotropic Brownian diffusion (Supplementary Fig. 1).

We have changed “large chirality” to “small pitch” and added more explanation of the omnidirectional movement in the section of “**Dynamics of directrons**”.

Q 4.6 In Figure 2(e) the authors claim that "Pair interaction potential function(extracted from the radial distribution function $g(r)$ shown in the inset) of the soliton...." How was the pair interaction calculated from $g(r)$? Please give full explanation of this important part of the manuscript.

A 4.6 We thank the reviewer for his/her comments. The pair interaction potential function is calculated according to ref [97], “when the radial distribution function ($g(r)$) is determined in the limit of infinite dilution, the pair interaction potential can be evaluated through the Boltzmann distribution $U(r) = -k_B T \ln[g(r)]$.” Such a calculation has also been used in other studies, such as ref [25]. We measured the radial distribution function of the solitons at very low density (low applied voltage), and then calculated the pair interaction potential through the Boltzmann distribution. We have added more details about the calculation in the Methods section.

Q 4.7 Figure 5. needs more accurate description. Is it correct to understand that it represents a probability $P(n)$ that a flock of solitons will have n members? If so, please write it clearly, using full and understandable sentences.

A 4.7 We thank the referee for his/her suggestion and are sorry for our unclear formulation. We have changed the caption of Fig. 5 as “**Fig. 5 Frequency distribution of directron group sizes (number of individual directrons, n , in each group)**. It demonstrates the probability of finding a soliton group composed of n directrons. The red line

Q 4.8 The authors show circular motion of solitons in Figure 11, but it is not clearly stated that the liquid crystal is the same as for previous experiments, but has a slightly larger pitch. Flocks of solitons experiencing circular motion is observed, but it is not clear if these solitons are of the same sort as those studied in planar cells? If so, why do we see similar solitons at lower chirality, i.e. larger pitch, while the authors have claimed that flocks are observable for large chirality?

A 4.8 We thank the referee for his/her comments. Yes, the LC material and the solitons are the same compared to previous experiments, except that the pitch is changed. We have added more details about the experimental setup in the “**Circular collective motion of solitons commanded by topological defects**” part. We have also added more explanations about the solitons to demonstrate that the solitons observed here are the same type as those observed in homogeneous cells.

Actually, the solitons can also be generated in homeotropic cells with chiral nematics of pitch $p = 2 \mu\text{m}$, i.e. the same LC used in homogeneous cells. The reason that we use LCs with larger pitch ($p = 10 \mu\text{m}$) here is because we cannot induce the individual umbilic defect pairs in LCs with small pitches. The defects in LCs with small pitches are very complicated. There are also linear disclinations generated throughout the sample which trap and hinder the motion of solitons, as the reviewer can see from the figure below.

The reason that the flocks cannot be observed in LCs of large pitches in homogeneous cells is because the solitons in those cases move either parallel or perpendicular to the alignment direction. They can hardly change their moving directions during motion and collisions. The reason that we

can see the flocks in homeotropic cells filled with LCs of large pitches maybe can be attributed to the rotational symmetry of the system, i.e., there is no specific alignment direction. The solitons can spontaneously choose a random moving direction and can freely change their moving directions through collisions with other solitons and interactions with topological defects. The solitons cannot be generated in homeotropic cells filled with achiral nematics.

Q 4.9 The authors mention that solitons could be used for "micro cargo transport". I would be convinced if the authors showed how solitons are able to move a microparticle in a LC.

A 4.9 We thank the reviewer for his/her interest in micro-cargo transport by solitons. Actually, the micro-cargo transport by dissipative solitons has been realized and reported by us and other groups previously (refs [46, 47, 50, 92]). The reviewer can also find movies about the micro-cargo transport in those studies. We also plan to realize more complicated micro-cargo transport by using these solitons in our future studies. We believe the present manuscript would be overloaded if adding more content on micro-cargo transport which is not quite the topic of this paper. We kindly suggest for now for the reviewer to read refs [46, 47, 50, 92] if he/she is interested in this topic.

Q 4.10 The article is not well written, the structure is not clear, in some places the same issues are repeated several times. The narration of the article is poor. The article is descriptive with limited aim to explain the physics behind the observed phenomena.

A 4.10 We again thank the reviewer for evaluating our work and his/her precious comments and suggestions which are very helpful to us to improve our manuscript. We have rectified the manuscript based on the comments and suggestions given by all the reviewers. We hope that this has improved the structure of the manuscript as well as its content, the explanations and the discussions.

REVIEWER COMMENTS

Reviewer #1 (Remarks to the Author):

I cannot recommend this manuscript for publication in this revised form. In response to my remarks, authors offered extensive discussions of what equipment they have in the lab and how they cannot do much more beyond what they already did, but most of my concerns remain not addressed. In the response authors confirmed that they do not understand the structure or physical origins of the solitons they study. They changed the name of solitons, but it is not clear if their solitons have much in common with the ones whose name they now adopt.

Reviewer #2 (Remarks to the Author):

**The revised manuscript addresses my comments and I recommend publication.
Oleg D. Lavrentovich**

Reviewer #3 (Remarks to the Author):

Basically, the authors mentioned most of my concerns but the description on the possible structure of the soliton. In my opinion, the most unique point of the present work is unprecedented dynamics of the soliton and the scaling-free-like behavior common with other active matter systems. In this sense, the work and presented physical behaviors are already enough unique and interesting, while the solitonic structure and physical analysis in the work are not very clear. But it is also true that for the broad readership in the wide field of science, the paper requires to present (propose) what are the solitons which they are observing in the present study, at least in cartoon.

As for dynamics, I'm sure that in future, efforts by many theoretical physicists will figure out such a complex phenomenon, but certainly it is not easy. But the presentation by experimentalists should be made carefully to give sufficient information for such theoretical physicists to think about the problem. Therefore, in this point I agree with some comments by other reviewers. Although I believe that the present system is a chiral version of the nematic system presented by Li et al. in Nat. Commun. in 2017 and in this case the present solitons are not topologically unique, still there is a possibility that chirality may induce a topologically unique state in the present solitons. So it looks good idea to include some more structural analysis before publication. Maybe fluorescent confocal microscopy may give useful information to consider the structure. So I recommend the authors to do this.

Reviewer #4 (Remarks to the Author):

The authors have considered all my previous remarks, comments and recommendations. They have substantially edited the manuscript. The narration of the manuscript has been improved, although I am not quite happy with it. However, in view of the novelty of this work and the importance of their findings, I can recommend the manuscript for acceptance to Nature Communications.

We again thank you and the reviewers for a thorough evaluation of our revised work. Below we list all the reviewers' comments and our replies. All the changes throughout the manuscript are marked with track changes. The changes in the last revision are marked with yellow color throughout the manuscript.

Reviewer #1 (Remarks to the Author):

Q 1.1 I cannot recommend this manuscript for publication in this revised form. In response to my remarks, authors offered extensive discussions of what equipment they have in the lab and how they cannot do much more beyond what they already did, but most of my concerns remain not addressed. In the response authors confirmed that they do not understand the structure or physical origins of the solitons they study. They changed the name of solitons, but it is not clear if their solitons have much in common with the ones whose name they now adopt.

A 1.1 We thank the reviewer for his comments. However, we think we have answered the reviewer's comments point by point in the last review report. The main point of the reviewer's comments in the last report is that the solitons in our experiment are torons or skyrmions. To address that, we have done more characterization of the solitons with samples of different pitches and cell gaps and given a list of reasons to prove that the solitons in our experiment are not torons or skyrmions. The reviewer was also concerning that our work is similar to the work of Sohn's. To address that, we have demonstrated the differences between our work and Sohn's work and the novelty and the importance of our work. We have also discussed and excluded the possibility that the solitons are driven into motions by fluid flows and demonstrated that the defects in Fig. 12 are not dowser texture but umbilic defects, etc.

Although we did not totally understand the formation mechanism and the director structure of the solitons, we have provided a reasonable explanation of the formation mechanism of the solitons (the first paragraph of the results section) and deduced the structure of the solitons from the POM images of the solitons and their formation mechanism (Supplementary Figure 2 which has now been moved to the body of the main manuscript as Fig. 3). We hope that this clarifies the structure for the readers, including the broader audience, right from the paper without the need to refer to the Supplementary Information.

We changed the name of the solitons to "directrons" as suggested by reviewer #2 because the term "soliton" is rather broad and can in our case be narrowed down and more concisely be referred to as directron. The experimental system and the materials, (except the pitch of the LCs which is smaller in this case), are the same as the ones in our previous publication (Ref 31) and the experimental conditions (the physical property of the LC materials, the anchoring condition of the cells, the cell gaps, the amplitude and frequency of the applied voltages, etc.) are basically the same as the ones in other publications of directrons (Refs 29, 30, 34, 35). The frequency dependent stability of the solitons in our experiment is also consistent with the directrons (Refs 29-31, 34, 35). The structure of the solitons in our experiment deforms periodically with the frequency of the applied electric field which is also consistent with the behavior of the directrons (Refs 29-31, 34, 35). Further, the nucleation and formation of the solitons in our experiment are also similar to the directrons which usually nucleate at dust particles and electrode boundaries (Refs 29-31, 34, 35).

Most importantly, the formation of the solitons in our experiment is closely related to the conductivity of the LC media. If the conductivity of the LC media is only slightly changed, such as by doping a very small amount of fluorescent dye, the solitons will not be generated and instead a global electro-convective pattern will emerge. Such a behavior is also consistent with the formation of directrons (Refs 29-31, 34, 35). The points mentioned above give us the confidence to be sure that the solitons in our experiment are in fact directrons.

Reviewer #2 (Remarks to the Author):

Q 2.1 The revised manuscript addresses my comments and I recommend publication.
Oleg D. Lavrentovich

A 2.1 We thank the reviewer for recommending the publication of our work.

Reviewer #3 (Remarks to the Author):

Q 3.1 Basically, the authors mentioned most of my concerns but the description on the possible structure of the soliton. In my opinion, the most unique point of the present work is unprecedented dynamics of the soliton and the scaling-free-like behavior common with other active matter systems. In this sense, the work and presented physical behaviors are already enough unique and interesting, while the solitonic structure and physical analysis in the work are not very clear. But it is also true that for the broad readership in the wide field of science, the paper requires to present (propose) what are the solitons which they are observing in the present study, at least in cartoon.

As for dynamics, I'm sure that in future, efforts by many theoretical physicists will figure out such a complex phenomenon, but certainly it is not easy. But the presentation by experimentalists should be made carefully to give sufficient information for such theoretical physicists to think about the problem. Therefore, in this point I agree with some comments by other reviewers. Although I believe that the present system is a chiral version of the nematic system presented by Li et al. in Nat. Commun. in 2017 and in this case the present solitons are not topologically unique, still there is a possibility that chirality may induce a topologically unique state in the present solitons. So it looks good idea to include some more structural analysis before publication. Maybe fluorescent confocal microscopy may give useful information to consider the structure. So I recommend the authors to do this.

A 3.1 We thank the reviewer for his comments. Actually we have proposed the director structure of the directrons from the POM images and the formation mechanisms of the directrons in the last version of the manuscript as the Supplementary Figure 2. It was indeed a bit unfortunate to place this figure in the Supplementary Information, so, in order to make the manuscript more transparent and concise for the broad readership, we now have moved the Figure to the main body of the manuscript as Figure 3. We have also described the formation mechanism and the structure of the solitons in detail in the first and second section of the results section.

We thank the reviewer for his suggestion. However, most importantly, as we now mentioned in the manuscript and the last response to the reviewers report, the formation of the directrons is very

highly dependent on the conductivity of the LC medium. Doping a fluorescent dye, even a very small amount, into the LC systems will largely change the conductivity of the LC. To investigate the fluid flows as suggested by reviewer #1 (A 1.3 in the last review report and *Methods* in the manuscript), we doped a small amount of fluorescent dye (~0.04 wt%) into the LC system. As a result, no directron was generated, instead a global 2D grid convective pattern was formed in the doped system. Considering the topological property of the directrons, these show a frequency dependent stability. For instance, the directrons are stable at $U = 30$ V, $f = 100$ Hz (Fig. 11), but disappear at $U = 30$ V, $f = 300$ Hz. If the directrons are topologically protected, such a frequency dependent stability does not intuitively make sense since changing the frequency of the applied electric field does not destroy the topological structure of the solitons.

Reviewer #4 (Remarks to the Author):

Q 4.1 The authors have considered all my previous remarks, comments and recommendations. They have substantially edited the manuscript. The narration of the manuscript has been improved, although I am not quite happy with it. However, in view of the novelty of this work and the importance of their findings, I can recommend the manuscript for acceptance to Nature Communications.

A 4.1 We thank the reviewer for recommending the publication of our work.

REVIEWERS' COMMENTS

Reviewer #1 (Remarks to the Author):

I cannot recommend this manuscript and I would not recommend the other manuscripts that these authors somehow managed to publish in more archival-type journals. Rather than doing more experiments and testing models and possibilities, authors try to "talk their way through" the review process. I felt it was too obvious that the 1st revision was not adequate, but perhaps this was hidden in many pages of the responses to the previous reports. In relation to my first report, the following points were not addressed (referring to the 1st response letter):

(Q1.1-A1.1) Authors agree that they do not understand the physics behind stability of their "solitons" by saying, I quote (page 8 of initial response): "We agree with the reviewer's and Pikin's opinion that "the underlying mechanism of the formation of the 'dissipative solitons' is still not very clear and requires further experimental and theoretical investigations.""

They even discuss what experiments could be done to rule out charge injection as a possible mechanism, saying "... To totally exclude the possibility of charge injection, further experiments are required, for instance, one can make a cell with "blocking" electrodes (e.g. covering the electrodes with thin layers of pyrex glass)...", but they do not do these experiments or anything else except talking their way through with some hypothetical possibilities not proved by experiments or rigorous models.

In responding to my suggestion to use high-NA objective to get more clear images with higher magnification, authors exposed their problem of using thick glass with high NA objectives. Why would not they make LC cells with thin glass, as it would be appropriate for objectives with NA like 1.4? Of course thick glass substrates cannot work with high NA objectives due to short working distance, but to overcome this "challenge" they would just need to use thin glass, not the commercial cells that they use as a black box. It is interesting to read "...the only available equipment to characterize the structure of solitons in our lab is the polarizing microscope...". "...and the light intensity of our microscope is not strong enough..." Well, authors could visit labs of other researchers in their department or in the UK – this is not an appropriate justification for poor understanding of the system authors want to publish about. How about doing computer simulations and just comparing experiments and polarizing microscopy videos? If authors were to know the underlying physics and structure of their objects under study, this would be easy and convincing. What these objects are and what they are not remains an open question, and surely there are 2 different types of "soliton" objects (not one, because of coexistence of one of them with cholesteric fingers in homeotropic cells and another in planar cells). In short, my main concern that authors do not know the structure of solitons and physics describing why they exist is not addressed.

(Q1.2-A1.2) Here my main concern was that authors were drawing completely impossible structure in the original version's Figure 1d (also criticized by other reviewers and since then removed). The response was non-scientific again (page 10, 2nd paragraph): "The solitons here may look like torons or skyrmions to some extent, however, we doubt that the solitons in either planar cells or homeotropic cells are torons..." Clearly, authors do not know with certainty what they work with. I asked them to do experiments, but all they do is write irrelevant things, repeating themselves as I read again "...light intensity of our microscope is not strong enough, the images are not very clear..." The 2nd revision has an updated "model" in Fig. 3... It is clearly impossible that such a structure would occur in a homeotropic cell with pitch comparable to cell thickness. I see no energetic or other reasons for such structure to be stable. Authors seem to say they have no confocal microscopes and cannot dope dyes as this destroys their samples, thus they cannot study 3D structure by confocal m-py. However, in changing the name from butterfly to "directron" authors imply that these are somehow similar solitons to "directrons" which were actually studied by confocal m-py in dye-doped samples. This is inconsistent then - authors contradict themselves. Responses and text also mention that the studied solitons differ from directrons studied by others (cannot pass through...) – again a contradiction and evidence that authors deal with something totally different. Circular vs asymmetric shape - another difference and contradiction as compared to what authors now claim could be "directrons".

(Q1.3-A1.3) I asked to check the nature of motion and if the mass transport is present during "soliton motion", but there was nothing conclusive in the response letter or revisions. They again write a lot, but no conclusive experiments.

(Q1.4-A1.4) I was ok with how this was reworded, better than in the original version

(Q1.5-A1.5) I still see no reason not to use the name given by Cladis in 1997. Changing the name does not add to novelty. The statement "... However, unlike the reviewer suggested, in fact, we do not have more tools to uncover the structure of solitons compared to what Cladis et al had available. The fact that the structure of the solitons has been unclear for about 30 years also..." Once again authors confess to not understanding the structure, but (even if not in their lab) dozens of different techniques could be used to uncover this structure. For example, AFM and other types of imaging after polymerization could be used, as well as CARS, FCPM, multiphoton absorption based fluorescence imaging, birefringence-based tomographic reconstruction of the director field, numerical modeling and comparison with high resolution POM images... Other referees were suggesting this too, but authors just try to talk their way through

(Q1.6-A1.6) Authors agreed with my remarks about the potential role of elastic interactions, but they did not characterize such interactions. Moreover, quadrupolar nature of such interactions would strongly disagree with the observations of motions and model that authors try to guess. What exactly could the short- and long- range forces associated with that authors write about?

(Q1.7-A1.7) I am ok with the response to this question

(Q1.8-A1.8) Authors did not do experiments showing clearly what they deal with and revealing unambiguously that the "solitons" they observe are or are not directrons, or butterflies, or bullets, or torons, or bubbles, or skyrmions, or finger loops, or helions, or droplets, or hopfions, or some contamination or phase separated dopant or something else. They try to say here that they do the same characterizations as in the past works by Sohn but with a different type of solitons, which is why this is novel. So far I have not seen data revealing the structure and there is not enough information to assess this claim.

(Q1.9-A1.9) As before, I cannot recommend publication because most concerns have not been addressed. Like one of the other referees, I believe authors "mentioned" most of my remarks and suggestions, but they did not address/follow them.

In the 2nd response letter and revision authors had another chance but did not take it. Without further experimenting, they draw a model that is inconsistent with perpendicular boundary conditions and sample geometry in homeotropic cells. They refer to several articles in other journals claiming they have the same thing. Then they contradict themselves as they also wrote about differences between their objects and the ones in other papers (circular versus asymmetric, passing through each other versus not, stable with respect to doping dyes and suitable for 3D confocal imaging versus not). Of course, this also poses a question of novelty. If so much was published already (still without understanding the system, and even if the collections of "solitons" were smaller in these other articles), is this work suitable for Nature Communications? At this point I recommend that the article is rejected, authors do more experiments to boost their understanding and only then submit their work to a more specialized journal.

Reviewer #3 (Remarks to the Author):

The authors properly mentioned most of my concerns except the internal structure of the solitons. Of course, the phenomenon described therein is interesting. Now I think that this result should be opened

and further discussed widely in the scientific community. So the present manuscript is recommended for publication in Nature Communications.

Reviewer #1 (Remarks to the Author):

Q 1.1 I cannot recommend this manuscript and I would not recommend the other manuscripts that these authors somehow managed to publish in more archival-type journals. Rather than doing more experiments and testing models and possibilities, authors try to “talk their way through” the review process. I felt it was too obvious that the 1st revision was not adequate, but perhaps this was hidden in many pages of the responses to the previous reports. In relation to my first report, the following points were not addressed (referring to the 1st response letter):

(Q1.1-A1.1) Authors agree that they do not understand the physics behind stability of their “solitons” by saying, I quote (page 8 of initial response): “We agree with the reviewer’ s and Pikin’ s opinion that “the underlying mechanism of the formation of the ‘dissipative solitons’ is still not very clear and requires further experimental and theoretical investigations.””

They even discuss what experiments could be done to rule out charge injection as a possible mechanism, saying “... To totally exclude the possibility of charge injection, further experiments are required, for instance, one can make a cell with “blocking” electrodes (e.g. covering the electrodes with thin layers of pyrex glass)...” , but they do not do these experiments or anything else except talking their way through with some hypothetical possibilities not proved by experiments or rigorous models.

In responding to my suggestion to use high-NA objective to get more clear images with higher magnification, authors exposed their problem of using thick glass with high NA objectives. Why would not they make LC cells with thin glass, as it would be appropriate for objectives with NA like 1.4? Of course thick glass substrates cannot work with high NA objectives due to short working distance, but to overcome this “challenge” they would just need to use thin glass, not the commercial cells that they use as a black box. It is interesting to read “...the only available equipment to characterize the structure of solitons in our lab is the polarizing microscope...” . “... and the light intensity of our microscope is not strong enough...” Well, authors could visit labs of other researchers in their department or in the UK – this is not an appropriate justification for poor understanding of the system authors want to publish about. How about doing computer simulations and just comparing experiments and polarizing microscopy videos? If authors were to know the underlying physics and structure of their objects under study, this would be easy and convincing. What these objects are and what they are not remains an open question, and surely there are 2 different types of "soliton" objects (not one, because of coexistence of one of them with cholesteric fingers in homeotropic cells and another in planar cells). In short, my main concern that authors do not know the structure of solitons and physics describing why they exist is not addressed.

A 1.1 We thank the reviewer for the comments. Yes, we said that the formation mechanism of the directrons is not completely understood. It is because there are too many factors that may contribute to the formation of the directrons, such as motion and injection of ions, electro-convections, flexo-electric effect, the dielectric properties and conductivity of the LC material, surface anchoring, etc., and all these factors are coupling with each other. The formation mechanism is complex and requires the efforts from not only experiments, but also theories and numerical simulations. That is why there are increasing numbers of theoretical works (S. A. Pikin, Journal of Experimental and Theoretical Physics, 2021, 132, 637; S. A. Pikin, Journal of Surface Investigation, 2019, 13, 1078; A. N. Earls,

Flexoelectricity and 3D solitons in nematic liquid crystals, Doctoral dissertation, University of Minnesota, 2019; Y. Garnovskiy, Nano Express, 2021, 2, 012004; D. Kumar, et al., Nonlinear Dynamics, 2022, 107, 2717; M. Higazy, et al., Journal of Ocean Engineering and Science, 2022, doi.org/10.1016/j.joes.2022.01.007; A. Earls and M. C. Calderer, Liquid Crystals, 2022, doi.org/10.1080/02678292.2021.2006812; etc.) being published recently and trying to explain the phenomenon. On the other hand, we have given a reasonable qualitative explanation about the formation of the directrons in the section on the **formation of directrons** and in the first response letter (A 1.1). The objective of this work is not the formation mechanism of the directrons and we do not think that this can be done only by one group in one investigation.

As suggested by the reviewer, we had characterized our solitons with a high-NA objective (Figure 3 in the last version of manuscript which now has been moved to Figure 1 (d)). We also made LC cells with thin glass substrates (cover slips), however this did not really help in finding an explanation for the above question as one can see from the figure below.

On the other hand, although the images obtained through the high-NA objective is not very clear, one can still distinguish the quadrupolar symmetric texture of the directrons (Figure 1 (d)). We thus deduced the director configuration of the directrons through those POM images. The reviewer also suggests visiting other labs in the UK to do further characterization, however, we cannot see what other kinds of characterization can be really helpful. The main problem is, as we mentioned in the first response letter, that fluorescent microscopy does not work because doping the fluorescent dye into our sample will suppress the generation of directrons by changing the dielectric and conductive conditions. The reviewer mentioned to polymerize the sample and do AFM and other types of imaging. However, as we mentioned before, the formation of the directrons is closely dependent on the physical properties of the LC media, doping different kinds of monomers into our LC system completely changes the conductivity and elastic constants of the LC system and thus suppresses the formation of directrons.

The solitons in the homogeneous cells and the homeotropic cells are both directrons. Unlike the reviewer stated, the ones in the homeotropic cell cannot coexist with cholesteric fingers because they appear after the Fredericksz transitions and the cholesteric fingers have already disappeared at such a high voltage. We have mentioned this in the first response letter (A 1.2). We do not know

where the reviewer arrives at the conclusion “*because of coexistence of one of them with cholesteric fingers in homeotropic cells*”. If he/she concludes this from **Figure 13** (which is now moved to **Figure 10**), then the fingerprint texture in the figure is obtained after removal of the applied electric field. If he/she arrives at the conclusion from the POM image in the first response letter (**A 1.2**), that region where torons exists is one which is not applied with an electric field.

We believe that we have provided a reasonable explanation of the formation of the directrons (in the section of **Formation of directrons**), the evidence to prove that these solitons are directrons (**A 1.2** in the first response letter), and the structure of the directrons (**Figure 1 (e) and (f)**).

Q 1.2 (Q1.2-A1.2) Here my main concern was that authors were drawing completely impossible structure in the original version’s Figure 1d (also criticized by other reviewers and since then removed). The response was non-scientific again (page 10, 2nd paragraph): “The solitons here may look like torons or skyrmions to some extent, however, we doubt that the solitons in either planar cells or homeotropic cells are torons...” Clearly, authors do not know with certainty what they work with. I asked them to do experiments, but all they do is write irrelevant things, repeating themselves as I read again “...light intensity of our microscope is not strong enough, the images are not very clear...” **The 2nd revision has an updated “model” in Fig. 3... It is clearly impossible that such a structure would occur in a homeotropic cell with pitch comparable to cell thickness.** I see no energetic or other reasons for such structure to be stable.

Authors seem to say they have no confocal microscopes and cannot dope dyes as this destroys their samples, thus they cannot study 3D structure by confocal m-py. However, in changing the name from butterfly to “directron” authors imply that these are somehow similar solitons to “directrons” which were actually studied by confocal m-py in dye-doped samples. This is inconsistent then - authors contradict themselves. Responses and text also mention that the studied solitons differ from directrons studied by others (cannot pass through...) – again a contradiction and evidence that authors deal with something totally different. Circular vs asymmetric shape - another difference and contradiction as compared to what authors now claim could be “directrons”.

A 1.2 We thank the reviewer for his comments. Yet, contrary to his/her statement, we do know what we are working with and have pointed out that the solitons here are directrons not skyrmions or anything else. We have also provided evidence to prove that the solitons are directrons in the previous response letter (**A 1.2** in the first response letter), which may have been overread. The reviewer insists that the solitons are not directrons but torons and that we have no idea what they are. The definition of “directron” we use is: Directrons are director perturbations which are topological trivial particle-like localized dissipative solitary waves (B. X. Li, et al., Phys. Rev. R., 2020, 2, 013178). We have explained that the solitons in our system are not topological solitons (**A 1.2** in the first response letter). The solitons are powered by the electric field and disappear if the applied voltage is lower than a threshold value which demonstrates that they are dissipative solitons. The solitons are particle-like localized deformation of director field. Although the director structure of the solitons in our system may be slightly different from each other due to different pitches and different surface anchoring, their texture, formation, stability and dynamics are similar and are all equivalent to those of directrons.

The reviewer does not agree with the director structure we provided but does not give a specific explanation by just saying that “*It is clearly impossible that such a structure would occur in a*

homeotropic cell with pitch comparable to cell thickness. I see no energetic or other reasons for such structure to be stable.” We need to remember that the dielectric anisotropy of our LC system is negative, thus by applying an electric field perpendicular to the cell substrates transforms the LC bulk from the homeotropic structure to the translationally invariant configuration or quasi-planar state (there might be homeotropic layers very close to the alignment layer on the glass surfaces, but such layers are very thin). This quasi-planar state under electric field application is very much compatible to the directron structure that we proposed.

Again as we mentioned in the manuscript and the previous response letters, the formation of directrons is closely dependent on the conductivity of the LC system. The LC media in which the confocal microscopy of directrons is made (CCN-47) is different from ours here (ZLI-2806). This has also been demonstrated by S. Aya et al. (*Nature communications* **11**, 1-10 (2020)): to generate directrons in a (-+) system, the conductivity, σ , has to be $8 \times 10^{-9} < \sigma < 4 \times 10^{-8} \Omega^{-1} \text{m}^{-1}$. The conductivity anisotropy of CCN-47 is about $-1.2 \times 10^{-9} \Omega^{-1} \text{m}^{-1}$, while the one of ZLI-2806 is about $1.3 \times 10^{-8} \Omega^{-1} \text{m}^{-1}$. In addition, the formation of directrons is also dependent on ions. The ion concentration of CCN-47 in that publication is also different from ours. As a result, the fact that doping a tiny amount of fluorescent dye into CCN-47 does not suppress the generation of directrons does not imply that this method also works with other liquid crystals such as ZLI-2806 as it is used here. In addition, the authors in that publication (B. X. Li, et al., *Nat. Com.*, 2018, 9, 2912) also mentioned that by doping tetrabutylammonium bromide into CCN-47, the conductivity is increased to $10^{-7} \Omega^{-1} \text{m}^{-1}$ and the formation of directrons is suppressed. Therefore, in contrast to the reviewer’s suggestion, the disappearance of directrons in our system by doping with a fluorescent dye is consistent with the results and conclusions of the previous works and actually further demonstrates that the solitons observed here are not skyrmions or torons but directrons. The different behaviors (pass through vs collide, circular vs asymmetric shape) that the reviewer mentioned above is due to our LC system being a chiral LC system.

Q 1.3 (Q1.3-A1.3) I asked to check the nature of motion and if the mass transport is present during “soliton motion”, but there was nothing conclusive in the response letter or revisions. They again write a lot, but no conclusive experiments.

A 1.3 We believed to have explained this issue quite clearly in the first response letter (**A 1.3**) and in the manuscript. We first doped a fluorescent dye and also quantum dots into our system in order to observe if there is any fluid flow. However, both dopants lead to the suppression of the generation of directrons. We then doped microparticles into our system. By observing the motion of the microparticles, we did not see any evidence that demonstrated the existence of flows. We also excluded the possibility of the existence of backflow, electro-convection, and the flows generated by ions.

Q 1.4 (Q1.4-A1.4) I was ok with how this was reworded, better than in the original version

A 1.4 We thank the reviewer for the comments.

Q 1.5 (Q1.5-A1.5) I still see no reason not to use the name given by Cladis in 1997. Changing the name does not add to novelty. The statement “... However, unlike the reviewer suggested, in fact,

we do not have more tools to uncover the structure of solitons compared to what Cladis et al had available. The fact that the structure of the solitons has been unclear for about 30 years also...” Once again authors confess to not understanding the structure, but (even if not in their lab) dozens of different techniques could be used to uncover this structure. For example, AFM and other types of imaging after polymerization could be used, as well as CARS, FCPM, multiphoton absorption based fluorescence imaging, birefringence-based tomographic reconstruction of the director field, numerical modeling and comparison with high resolution POM images... Other referees were suggesting this too, but authors just try to talk their way through

A 1.5 We thank the reviewer for his comments. However, we do not see the reason why the reviewer insists for us to use the name (butterfly) given by Cladis in 1997 if “directron” can well describe the properties of the solitons in our system. We also did not change the name from “butterfly” to “directron”. In the last version of manuscript, we simply call our solitons “soliton”. We changed name to “directron” just because “soliton” is a very general term and not descriptive enough of our findings as suggested by the reviewer #2.

We have given the structure of the directron as requested by the reviewers (Figure 1). We have explained that polymerization, FCPM and multiphoton imaging etc. are not a good choice for characterizing the directrons in our system because the doping of either monomers or fluorescent dyes will change the conductivity of the LC media and thus suppress the generation of directrons. The CARS (Coherent anti-Stokes Raman spectroscopy) microscopy is a typical imaging technique for characterizing biological samples which depends on the characteristic vibrational energy states of different kinds of molecules. However, there is only one kind of LC molecules in our system. So we do not think this will be helpful.

Q 1.6 (Q1.6-A1.6) Authors agreed with my remarks about the potential role of elastic interactions, but they did not characterize such interactions. Moreover, quadrupolar nature of such interactions would strongly disagree with the observations of motions and model that authors try to guess. What exactly could the short- and long- range forces associated with that authors write about?

A 1.6 We thank the reviewer for his comments. We tried to characterize the elastic interaction between the directrons through laser tweezing experiments. However, after many attempts, we came to the conclusion that the directrons could not be trapped or moved by laser tweezer (at least by the laser tweezer (Thorlab, EDU-OT3/M) that we used). We do not know what else one can do to further quantitatively characterize the interactions between the directrons.

The reviewer thinks that the interactions between directrons are quadrupolar. However, we did not claim this in the manuscript. Yes, the POM image of the directrons show a quadrupolar symmetric shape, but it does not mean that the interactions between the directrons are quadrupolar. We think the interactions between the directrons are more likely to be isotropic due to the helical twisting configuration of the chiral LC system. The isotropic interaction also agrees with the circular director structure of the directrons that we proposed in Fig. 1 and the hexagonal arrangement of the directrons.

Both short- and long-range forces can be explained by the elastic interactions between directrons. The director distortions usually attract each other at long distances in order to minimize the free energy of the system. However, if they get too close to each other, their structures are distorted, thus

leading to the repulsive interaction. The long-range attractive interaction leads to the formation of different directron flocks and the short-range repulsive interaction makes the directrons behave like colloidal particles and leads to the ordered hexagonal arrangement of the directrons.

Q 1.7 I am ok with the response to this question

A 1.7 We thank the reviewer for the comments.

Q 1.8 Authors did not do experiments showing clearly what they deal with and revealing unambiguously that the “solitons” they observe are or are not directrons, or butterflies, or bullets, or torons, or bubbles, or skyrmions, or finger loops, or helions, or droplets, or hopfions, or some contamination or phase separated dopant or something else. They try to say here that they do the same characterizations as in the past works by Sohn but with a different type of solitons, which is why this is novel. So far I have not seen data **revealing the structure** and there is not enough information to assess this claim.

A 1.8 We believe that we have given plenty of explanations and evidence in the manuscript and the first response letter to demonstrate that the solitons here are directrons, such as the dependence on the conductivity of the LC system, the frequency dependent threshold voltage and stability, the nucleation of the solitons, the dynamic behavior of the solitons, the morphing of the soliton structure with the electric field frequency, etc. (**A 1.2** in the first response letter). We have also experimentally tested and demonstrated that many of the methods that the reviewer mentioned are not appropriate for our system because the doping of a fluorescent dye or a monomer will suppress the generation of directrons.

We have clearly demonstrated that the solitons investigated here cannot be topological solitons, such as torons, bubbles, skyrmions, hopfions, finger loops, etc. This is outlined in the manuscript (in the section of “**Circular collective motion of directrons commanded by topological defects**”) and in the first response letter (**A 1.2**). They also cannot be droplets, contamination or phase separated dopant because the solitons show the same structure and dynamic behaviors. Their number density is dependent on the applied voltage and they disappear immediately once the applied voltage is lower than the threshold voltage. The butterflies and bullets are practically the same entities as directrons, it is just that different researchers use the different names.

We have also demonstrated the novelty of our work compared to that of Sohn (in the first paragraph of the discussion section and **A 1.8** in the first response letter). We do not know why the reviewer thinks “*They try to say here that they do the same characterizations as in the past works by Sohn but with a different type of solitons, which is why this is novel.*” There are indeed some similar characterization methodologies as in Sohn’s work, such as velocity order parameter, giant number fluctuations (Nat. Com. 10, 4744 (2019)). However, these techniques are also broadly used in many other investigations of active systems simply because they are very adequate (A. Bricard, et al., Nature, 503, 95, 2013; Nat. Com., 6, 7470, 2015; V. Narayan, et al., Science, 317, 105, 2007; J. Deseigne, et al., Phys. Rev. Lett., 105, 098001, 2010; etc.). We further characterized our system with other methods, such as spatial and temporal correlation functions, which are not used in Sohn’s work but are broadly used in many other research works of active systems (J. Toner and Y. H. Tu, Phys. Rev E, 58, 4828, 1998; H. P. Zhang, et al., PNAS, 107, 13626-13630, 2013; V. Schaller and

A. R. Bausch, PNAS, 110, 4488-4493, 2013; X. Chen, et al., Phys. Rev. Lett., 108, 148101, 2012).

Q 1.9 As before, I cannot recommend publication because most concerns have not been addressed. Like one of the other referees, I believe authors “mentioned” most of my remarks and suggestions, but they did not address/follow them.

A 1.9 We thank the reviewer for the comments. We believe we have answered all the questions raised by the reviewer and have done the experiments that we can do. The reviewer insists us to do more characterizations such as FCPM, AFM, multiphoton imaging, etc. We have tried a range of such methods and we have demonstrated that these techniques are not appropriate for our samples.

Q 1.10 In the 2nd response letter and revision authors had another chance but did not take it. Without further experimenting, they draw a model that is inconsistent with perpendicular boundary conditions and sample geometry in homeotropic cells. They refer to several articles in other journals claiming they have the same thing. Then they contradict themselves as they also wrote about differences between their objects and the ones in other papers (circular versus asymmetric, passing through each other versus not, stable with respect to doping dyes and suitable for 3D confocal imaging versus not). Of course, this also poses a question of novelty. If so much was published already (still without understanding the system, and even if the collections of “solitons” were smaller in these other articles), is this work suitable for Nature Communications? At this point I recommend that the article is rejected, authors do more experiments to boost their understanding and only then submit their work to a more specialized journal.

A 1.10 We do not agree with the reviewer. As explained above (**A 1.2**), the LC media used here has a negative dielectric anisotropy. Although the surface anchoring is homeotropic, the LC bulk first transforms into the translationally invariant configuration by applying the electric field before the generation of the directrons. As a result, the model we provide is compatible with homeotropic anchoring. There might be slight deviations due to the relatively large cholesteric pitch in homeotropic cells, but the model is qualitatively reasonable.

We have further explained in detail why the solitons investigated here are directrons. We have also explained the reasons why the differences (circular versus asymmetric, passing through each other versus not, stable with respect to doping dyes and suitable for 3D confocal imaging versus not) between the directrons in this work and the ones in previous works occur (please see **A 1.2**).

In conclusion, we thank the reviewer for the thorough review of our work throughout the review process and a number of very useful points, but we do not agree with the reviewer’s reasoning for suggesting to reject the revised manuscript.

Reviewer #3 (Remarks to the Author):

Q 3.1 The authors properly mentioned most of my concerns except the internal structure of the solitons. Of course, the phenomenon described therein is interesting. Now I think that this result should be opened and further discussed widely in the scientific community. So the present

manuscript is recommended for publication in Nature Communications.

A 3.1 We thank the reviewer for recommending our work for publication in Nature Communications and hope that we have improved on clarity still with the changes made in the revised manuscript.